# Learning to Decode Against Compositional Hallucination in Video Multimodal Large Language Models

**Wenbin Xing** [1]   **Quanxing Zha** [2]   **Lizheng Zu** [3]   **Mengran Li** [1]   **Ming Li** [4]   **Junchi Yan** [5]

## Abstract

Current research on video hallucination mitigation primarily focuses on isolated error types, leaving *compositional* hallucinations—arising from incorrect reasoning over multiple interacting spatial and temporal factors largely underexplored. We introduce **OmniVCHall**, a benchmark designed to systematically evaluate both isolated and compositional hallucinations in Video Large Language Models (VLLMs). OmniVCHall spans diverse video domains, introduces a novel camera-based hallucination type, and defines a fine-grained taxonomy, together with adversarial answer options (*e.g.*, "All are correct" and "None of the above") to prevent shortcut reasoning. The evaluations of 39 representative VLLMs reveal that even advanced models (*e.g.*, Qwen3-VL and GPT-5) exhibit substantial performance degradation. We propose **TriCD**, a contrastive decoding framework with a triple-pathway calibration mechanism. An adaptive perturbation controller dynamically selects distracting operations to construct negative video variants, while a saliency-guided enhancement module adaptively reinforces grounded token-wise visual evidences. These components are optimized via reinforcement learning to encourage precise decision-making under compositional hallucination settings. Experimental results show that TriCD consistently improves performance across two representative backbones, achieving an average accuracy improvement of over 10%. The data and code are released at https://github.com/BMRETURN/OmniVCHall.

[1]School of Intelligent Systems Engineering, Sun Yat-sen University [2]Department of Computer Science, Huaqiao University [3]College of Electronic and Information Engineering, Shenzhen University [4]School of Artificial Intelligence, The Chinese University of Hong Kong, Shenzhen [5]Shanghai Jiao Tong University. Correspondence to: Ming Li <ming.li@u.nus.edu>.

*Proceedings of the 43rd International Conference on Machine Learning*, Seoul, South Korea. PMLR 306, 2026. Copyright 2026 by the author(s).

## 1. Introduction

The rapid evolution of video large language models (VLLMs) has enabled unprecedented capabilities in video understanding (Zou et al., 2024; Tang et al., 2025; Bai et al., 2025a; DeepMind, c). Despite this progress, the reliability of these models remains compromised by pervasive hallucinations, manifested as generated content that is inconsistent with the underlying visual evidence (Rawte et al., 2023; Sahoo et al., 2024). A growing body of work has investigated this issue and proposed methods to mitigate hallucinations in VLLMs (Bansal et al., 2024; Wang et al., 2024b; Zhang et al., 2024a; Kong et al., 2025; Rawal et al., 2025). They predominantly focus on single or isolated hallucination types, targeting specific failure modes such as action misinterpretation (Kong et al., 2025) or temporal inconsistencies (Choong et al., 2024; Li et al., 2025a).

However, in real-world settings, diverse hallucination types often emerge simultaneously and interact in a highly coupled manner (Li et al., 2023c; Maaz et al., 2024; Wang et al., 2024a;b; Bae et al., 2025; Yang et al., 2026; Chytas et al., 2025). This failure mode is referred as *compositional* hallucinations, which is particularly pronounced in VLLMs, as input videos typically contain rich and interdependent spatial and temporal cues. For example, object attributes, motion patterns, and camera dynamics are often jointly involved within a single query. Despite their prevalence, compositional hallucinations remain largely underexplored in existing cross-modal video studies.

To systematically address this gap, we introduce OmniVCHall, a comprehensive benchmark specifically designed to evaluate both isolated and compositional hallucinations. Unlike prior datasets that rely on coarse taxonomies, OmniVCHall defines eight fine-grained hallucination types, including a novel camera type that exposes observer-perspective vulnerabilities largely overlooked in existing studies. In addition, OmniVCHall incorporates adversarial options (*e.g.*, "All others are correct" and "None of the above") to discourage shortcut reasoning and enforce explicit grounding in the visual evidence. In total, OmniVCHall comprises 823 high-quality videos, drawn from both authentic real-world footage and carefully curated AI-generated content, paired with 9,027 visual question answer-

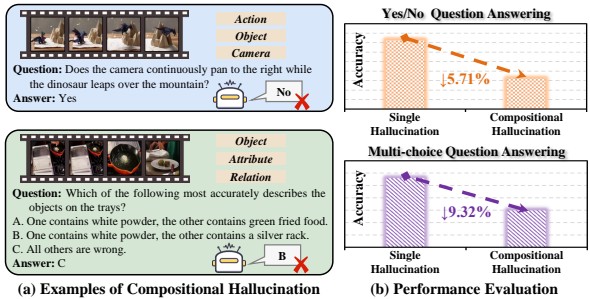

*Figure 1.* (a) Existing VLLMs frequently struggle with cross-modal video understanding tasks involving compositional hallucinations. (b) Extensive evaluation across 39 models reveals a pronounced accuracy drop (↓5.71% and ↓9.32%) when transitioning from single-factor hallucination queries to compositional ones.

ing (VQA) samples. Fig. 1 (a) provides two representative examples of VLLMs failing on yes/no and multiple-choice VQA tasks involving compositional hallucination. Fig. 1 (b) further illustrates that models achieve significantly lower accuracy on queries involving compositional hallucinations compared to those containing a single type. These results indicate that while VLLMs may handle isolated inconsistencies, their reasoning robustness collapses when confronted with multi-factor compositional contradictions.

To tackle this issue, we introduce TriCD, a triple-pathway video contrastive decoding calibration framework that unifies negative suppression via context-aware perturbation selection with saliency-aware spatiotemporal enhancement. Specifically, we propose an adaptive perturbation controller that dynamically selects context-aware video suppression operations, enabling flexible and instance-specific negative distraction. In parallel, we develop a saliency-guided enhancement module that strengthens visual evidence by fusing DINOv3 (Siméoni et al., 2025) spatial features with Farneback (Farnebäck, 2003) temporal motion cues, thereby improving visual grounding for complex spatiotemporal reasoning. Both modules are optimized via reinforcement learning to align the calibration process with the unique reasoning demands of each query. Extensive experimental results show that TriCD achieves substantial accuracy gains, *i.e.*, 9.61% for Qwen3-VL-Instruct-8B and 12.03% for VideoLLaMA3-7B, demonstrating its superiority as a robust and scalable solution for mitigating compositional hallucinations in VLLMs. Our contributions are:

- **Compositional Hallucinations in VLLMs.** We empirically show that queries involving multiple interacting hallucination factors pose substantially greater challenges than isolated hallucination types. To the best of our knowledge, this is the first systematic study of compositional hallucinations in VLLMs.

- **Benchmark.** We introduce OmniVCHall, a comprehen-

sive benchmark comprising high-quality videos, VQA pairs, a novel hallucination category, and adversarial answer options designed to prevent shortcut reasoning, enabling systematic evaluation of compositional hallucinations in existing VLLMs.

- **Method.** We propose TriCD, a triple-pathway contrastive decoding framework that combines adaptive negative suppression with saliency-aware spatiotemporal enhancement. Without updating the VLLM parameters, TriCD consistently improves performance in mitigating compositional hallucinations.

## 2. OmniVCHall Benchmark

Existing video hallucination benchmarks are often confined to narrow categories and predominantly focus on single-dimensional queries (Choong et al., 2024; Kong et al., 2025; Li et al., 2025a). However, in real-world scenarios, hallucinations rarely occur in isolation. They tend to emerge as entangled errors spanning multiple facets of video content. To bridge this gap, this section introduces OmniVCHall, a comprehensive benchmark designed to systematically evaluate both single and compositional hallucinations in VLLMs.

### 2.1. Definition

Let $G(V, T, A)$ be the minimal evidence-type set needed to support answer $A$ for question $T$ on video $V$. A case is single if $|G| = 1$, and compositional if $|G| \geq 2$ and removing any type makes the answer unsupported, ambiguous, or wrong. This follows the minimal-sufficiency view that a decision is grounded in the smallest sufficient set of evidence (I Amoukou & Brunel, 2022; Darwiche & Ji, 2022). For detailed analysis and experiments, please refer to Sec. B.1.

### 2.2. Taxonomy Design

As illustrated in Fig. 2 (a), OmniVCHall establishes a structured taxonomy that evaluates VLLMs across four critical perspectives, ensuring a rigorous assessment.

*Table 1.* Definition of eight hallucination types.

| Type | Definition |
|---|---|
| Object | Presence or identity of physical entities. |
| Scene | The overarching environment or setting. |
| Event | High-level semantic units or causal occurrences. |
| Action | Physical movement or behavioral patterns. |
| Relation | Spatial or logical interactions between entities. |
| Attribute | Static properties like color, size, or material. |
| Temporal | Chronological order or duration of occurrences. |
| Camera | Perception of cinematic lens dynamics. |

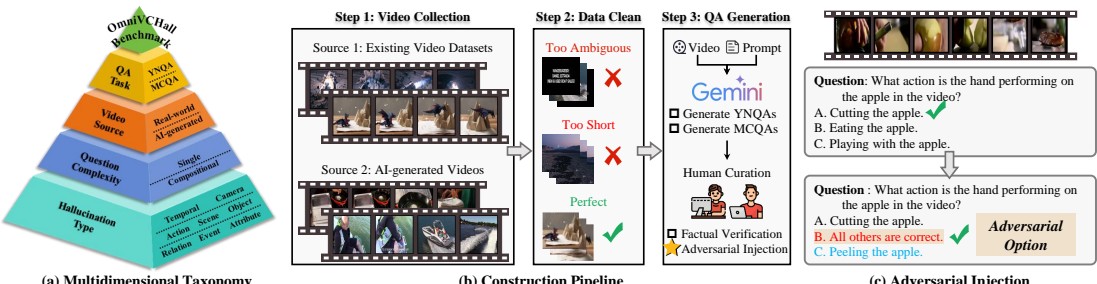

*Figure 2.* Overview of the OmniVCHall benchmark. (a) shows a hierarchical structure. (b) shows a three-step pipeline. (c) utilizes adversarial answer options (*e.g.*, "All are correct" and "None of the above") to discourage shortcut reasoning.

**Fine-grained Hallucination Type.** At the foundational level, we define eight fine-grained hallucination types covering distinct semantic facets of video understanding. Among them, we introduce a camera-related hallucination type to explicitly evaluate hallucination induced by cinematic dynamics, where camera operations such as zooming or panning are mistakenly interpreted as physical object motion. We formally define each type in Tab. 1 and provide representative examples in Sec. B.2. This fine-grained taxonomy enables a targeted analysis of model vulnerabilities.

**Hierarchical Question Complexity.** OmniVCHall incorporates two levels of task complexity. Single hallucination (S) tasks focus on an isolated error type, whereas compositional hallucination (C) tasks involve queries that span multiple hallucination types simultaneously, thereby exposing fundamental reasoning bottlenecks in current VLLMs.

**Diverse Video Source.** OmniVCHall comprises both real-world and AI-generated videos to ensure source diversity, capturing hallucination behaviors under both natural visual dynamics and synthetic content distributions.

**Multi-faceted QA Task.** To provide a comprehensive evaluation of model responses, OmniVCHall adopts two primary task formats: binary yes/no question answering (YNQA) for probing factual verification, and multi-choice question answering (MCQA) for assessing discriminative reasoning among competing semantic candidates. By intersecting these task formats with varying question complexity, we organize the evaluation samples into four sub-tasks:

- S_YNQA: Binary factual verification on a single type.
- C_YNQA: Binary factual verification on multiple types.
- S_MCQA: Discriminative reasoning on a single type.
- C_MCQA: Discriminative reasoning on multiple types.

### 2.3. Benchmark Construction

The construction of OmniVCHall follows a carefully designed three-step pipeline to ensure both data diversity and annotation reliability, as illustrated in Fig. 2 (b).

**Video Collection and Curation.** We initially curated a pool of 1,000 candidate videos, which were manually reviewed and filtered to ensure that each video provides clear and unambiguous visual evidence for reliable hallucination detection. This process yields a carefully curated corpus of 823 high-quality video clips from 11 heterogeneous domains, encompassing authentic real-world scenarios as well as synthetically generated videos with complex artifacts. Detailed source distributions are provided in Sec. B.3.

**QA Generation with Adversarial Injection.** We adopt a hybrid human-AI approach for QA generation. We first employ manual annotation to produce detailed ground-truth captions, accurately documenting the underlying spatiotemporal dynamics and object interactions. These serve as stable textual references, effectively reducing the risk of model-induced noise during subsequent QA construction. Secondly, Gemini-2.5-Pro (DeepMind, b) is utilized to generate initial candidate YNQA and MCQA questions based on the videos, human-annotated captions, and structured prompts. All generated questions are subsequently manually curated by expert annotators to verify factual correctness and to ensure that each compositional query is annotated with the corresponding hallucination types. A notable characteristic of OmniVCHall is the dynamic injection of adversarial options in MCQA tasks. Expert annotators manually incorporate options (*e.g.*, "All are correct" and "None of the above") to discourage reliance on simple elimination strategies or language-only priors. This encourages performing holistic cross-modal verification of each candidate before selecting an answer. Relevant cases are provided in Sec. B.5. To quantify the quality of the curated dataset and establish an empirical upper bound, we conduct a human performance evaluation involving three independent evaluators. As shown in Tab. 4, human evaluators achieve an average accuracy of 0.95, significantly outperforming current state-of-the-art VLLMs. Human performance remains consistently high and relatively stable across all sub-categories and task complexities. This indicates that, although adversarial options increase the difficulty for models, the tasks remain well-defined and solvable for human reasoners.

*Table 2.* Comparison of OmniVCHall with other video hallucination datasets. Ours provides the most comprehensive coverage across eight hallucination types, including the newly defined camera type, while uniquely balancing real-world and AI-generated sources.

| Benchmark | Hallucination Type | | | | | | | | QA | | | Video | | |
|---|---|---|---|---|---|---|---|---|---|---|---|---|---|---|
| | Object | Scene | Event | Action | Relation | Attribute | Temporal | Camera | YN | MC | Count | Real | Generated | Count |
| TempCompass (Liu et al., 2024) | ✗ | ✗ | ✓ | ✓ | ✓ | ✓ | ✓ | ✗ | ✓ | ✓ | 7,540 | ✓ | ✗ | 410 |
| VidHal (Choong et al., 2024) | ✓ | ✗ | ✗ | ✓ | ✓ | ✓ | ✓ | ✗ | ✗ | ✓ | 1,000 | ✓ | ✗ | 1,000 |
| VideoHallucer (Wang et al., 2024b) | ✓ | ✗ | ✓ | ✗ | ✓ | ✗ | ✓ | ✗ | ✓ | ✗ | 1,800 | ✓ | ✗ | 948 |
| EventHallusion (Zhang et al., 2024a) | ✗ | ✗ | ✓ | ✗ | ✗ | ✗ | ✗ | ✗ | ✗ | ✗ | 711 | ✓ | ✗ | 400 |
| MHBench (Kong et al., 2025) | ✗ | ✗ | ✗ | ✓ | ✗ | ✗ | ✗ | ✗ | ✓ | ✓ | 3,600 | ✓ | ✗ | 1,200 |
| VidHalluc (Li et al., 2025a) | ✗ | ✓ | ✗ | ✓ | ✗ | ✗ | ✓ | ✗ | ✓ | ✓ | 9,295 | ✓ | ✗ | 5,002 |
| ELV-Halluc (Lu et al., 2025) | ✓ | ✗ | ✓ | ✓ | ✗ | ✗ | ✓ | ✗ | ✓ | ✗ | 4,800 | ✓ | ✗ | 200 |
| **OmniVCHall (Ours)** | ✓ | ✓ | ✓ | ✓ | ✓ | ✓ | ✓ | ✓ | ✓ | ✓ | 9,027 | ✓ | ✓ | 823 |

## 2.4. Data Statistics

As illustrated in Fig. 3, we conduct a quantitative analysis of OmniVCHall across multiple dimensions to demonstrate its diversity and complexity. The benchmark contains 823 videos, balanced between real-world (423) and AI-generated (400) domains, resulting in a total of 9,027 queries. Temporally, most videos span 2 to 10 seconds, ensuring sufficient dynamic content for hallucination assessment. A key distinction of OmniVCHall lies in the higher linguistic complexity of compositional tasks. Compositional-type queries contain significantly more words than single-type queries, indicating the sophisticated reasoning required to disentangle multiple hallucination types. Tab. 3 presents the distribution of adversarial options in OmniVCHall, which is designed to mitigate shortcut reasoning by requiring models to perform explicit cross-modal verification for each candidate. The query distributions across different hallucination types are detailed in Sec. B.6.

Inter-Annotator Agreement (IAA) (Landis & Koch, 1977) analysis confirms the high reliability of OmniVCHall, with Cohen's Kappa scores of 0.84 for video description and 0.81 for QA validity (see in Sec. B.8).

Finally, as summarized in Tab. 2, OmniVCHall offers a more comprehensive evaluation of video-text misalignment by introducing the novel camera type hallucination and maintaining a balanced mix of real-world and AI-generated videos. This benchmark provides a valuable testbed for the community to develop more reliable and robust VLLMs.

## 3. TriCD Framework

Existing Contrastive Decoding (CD) methods rely on static perturbations (*e.g.*, shuffling) that fail to address the compositional hallucinations prevalent in complex video reasoning (Zhang et al., 2024a; Kong et al., 2025). Effective mitigation requires semantically adaptive negative samples, dynamically tailored to the video-query context. However, suppression alone is insufficient as it lacks explicit guidance toward

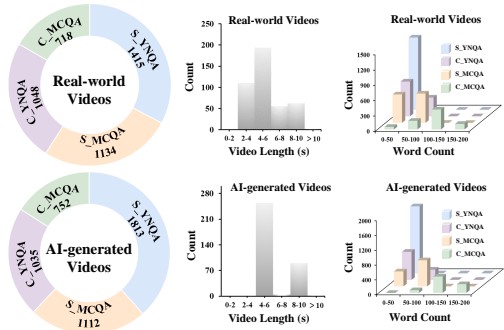

*Figure 3.* Statistical analysis of OmniVCHall. From left to right: distribution of the four sub-tasks across domains, histograms of video durations, and word count distributions highlighting the complexity of compositional tasks.

*Table 3.* Statistics of adversarial option injection in OmniVCHall. "AD" denotes queries with injected adversarial options.

| Type | Real-world | | AI-generated | |
|---|---|---|---|---|
| | S_MCQA | C_MCQA | S_MCQA | C_MCQA |
| AD | 315 | 243 | 268 | 179 |
| Total | 1134 | 718 | 1112 | 752 |
| Ratio | 27.78% | 33.84% | 24.10% | 23.80% |

critical visual evidence (Li et al., 2025a). To resolve this, we introduce TriCD, a three-step framework coupling an Adaptive Perturbation Controller (APC) with Saliency-Guided Enhancement (SGE), as shown in Fig. 4 (a).

### 3.1. Original Pass & State Extraction

TriCD begins by establishing a baseline understanding of the video-query context to guide subsequent adaptive perturbations and saliency enhancements. For an input video $V$ and textual query $T$, visual and textual tokens $(\mathbf{X}_v, \mathbf{X}_t)$ are concatenated for the LLM decoder. We extract hidden states $\mathbf{H}$ from the final transformer layer as a semantic anchor representing the grounded video-query context. For a VLLM (defined as $\theta$), the predicted original logit $q_t^o$ at $t$ is:

$$q_t^o = \text{logit}\theta(y_t \mid V, T, y_{<t}),\qquad(1)$$

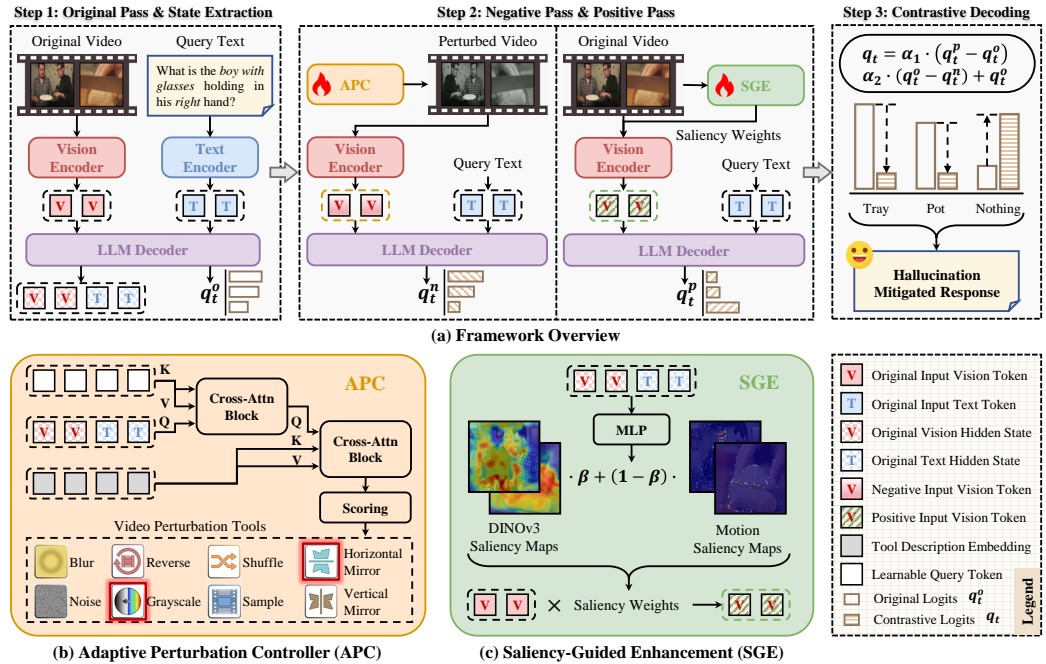

**Figure 4.** TriCD framework. (a) employs a three-step process that uses contrastive decoding to refine final predictions. (b) dynamically selects the most contextually relevant tools from a bank of eight video perturbation tools via cross-attention to construct a negative sample. (c) fuses spatial (DINOv3) and temporal (motion) saliency maps to reweight vision tokens, anchoring the positive pass to critical evidence.

where $y_{<t}$ are previous tokens. These original logits represent the initial, uncalibrated probability distribution over vocabulary $\mathcal{V}$ and remain susceptible to hallucinations.

### 3.2. Negative Pass & Positive Pass

TriCD executes two parallel branches to establish a calibrated inference boundary. The APC module generates context-aware negative samples to define what the model should avoid. The SGE branch anchors the model to evidence-rich regions to define what it should prioritize.

#### 3.2.1. ADAPTIVE PERTURBATION CONTROLLER

To address compositional hallucinations that static methods fail to capture, the APC module dynamically selects the most effective perturbation $\tau$ from a dictionary of eight tools $\mathcal{P}$. Each tool is designed to disrupt specific visual or temporal cues, providing a diverse set of negative samples for the contrastive decoding process. Their descriptions and examples are detailed in Sec. C.1 and Sec. C.2.

The APC module employs a dual-stage cross-attention mechanism to identify contextually relevant negative tools, as illustrated in Fig. 4 (b). Initially, learnable query tokens $\mathbf{Q}$ interact with the video-query hidden states $\mathbf{H}$ to encode specific semantic context. These conditioned queries then attend to the tool description embeddings $\mathbf{D}_{tool}$, which are derived by extracting the VLLM's hidden states after pro-

cessing text tokens that define the functionality of each perturbation tool. Specifically, for each tool, we construct the comprehensive analysis, as detailed in Sec. C.1.

Following the scoring head, tools are ranked in descending order, and all tools within a cumulative probability threshold $\gamma$ are selected to produce the perturbed sample $V^-$. Crucially, $V^-$ is designed to intentionally amplify the model's reliance on linguistic priors and hallucinations by corrupting critical spatiotemporal grounding. The resulting negative logit $q_t^n$ captures these induced hallucination pathways:

$$q_t^n = \text{logit}_\theta(y_t \mid V^-, T, y_{<t}), \qquad (2)$$

thereby serving as an informative "lower bound" that enables the subsequent contrastive decoding step to filter out misleading signals. Details are provided in Sec. C.3.

#### 3.2.2. SALIENCY-GUIDED ENHANCEMENT

Complementary to the negative pass in APC, the SGE module aims to sharpen the positive visual signal by anchoring the model's attention to the most informative spatiotemporal regions. SGE fuses object-centric spatial saliency with motion-aware temporal saliency via a learnable gating mechanism. Their examples are shown in Sec. C.2.

**Dual Saliency Extraction.** Following (Li et al., 2025a), we extract spatial saliency $\mathbf{s}_{spa}$ by computing the attention weights from the [CLS] token to all patch tokens in the last

layer of DINOv3 (Siméoni et al., 2025), effectively capturing foreground object importance. Simultaneously, we estimate temporal saliency $\mathbf{s}_{mot}$ using the Farneback algorithm (Farnebäck, 2003) to calculate optical flow. To isolate intentional actions from background jitter, we apply a dual-stage filtering process (Gaussian smoothing and temporal band-pass filtering) to the raw motion magnitude.

**Dynamic Fusion.** As illustrated in Fig. 4 (c), the final spatiotemporal signal is modulated by a learnable gating network. Conditioned on the context-rich hidden states $\mathbf{H}$, an MLP predicts a dynamic fusion weight $\beta \in [0, 1]$, yielding the unified weight: $\mathbf{w}_{sal} = \beta \cdot \mathbf{s}_{spa} + (1 - \beta) \cdot \mathbf{s}_{mot}$. The positive vision tokens $\mathbf{X}'_v$ are then enhanced via element-wise reweighting, $\mathbf{X}'_v = \mathbf{w}_{sal} \odot \mathbf{X}_v$, and fed into the decoder to obtain the positive logit:

$$q_t^p = \text{logit}_\theta(y_t \mid \mathbf{X}'_v, \mathbf{X}_t, y_{<t}). \tag{3}$$

This mechanism establishes a grounded "upper bound" by ensuring the model remains anchored to both critical objects and their dynamic interactions. Technical implementation details are provided in Section C.4.

### 3.3. Triple-Pathway Contrastive Decoding

With the original logits $q_t^o$, the adaptive negative logits $q_t^n$, and the saliency-guided positive logits $q_t^p$ successfully extracted, TriCD integrates them into a unified calibration framework. By treating the original prediction as a baseline and applying directed residual corrections, the model effectively widens the margin between grounded visual evidence and potential hallucination pathways. The final calibrated logit $q_t$ is formulated as below:

$$q_t = q_t^o + \alpha_1 \cdot (q_t^p - q_t^o) + \alpha_2 \cdot (q_t^o - q_t^n), \tag{4}$$

where both $\alpha_1$ and $\alpha_2$ are hyperparameters that adjusts the penalty strengt. The output prediction probability can be expressed as $y_t \sim \text{softmax}(q_t)$.

### 3.4. Optimization via Policy Gradient

We formulate the coordination of perturbation selection and saliency fusion as a Reinforcement Learning (RL) (Sutton et al., 1998; Li, 2017) problem to navigate the non-differentiable nature of discrete tool selection. By interacting with the frozen VLLM, the APC and SGE learn to maximize the suppression of hallucination pathways.

**Policy.** To facilitate the generation of negative samples, the APC is modeled via independent Bernoulli distributions, allowing for the simultaneous sampling of multiple perturbation tools. Conversely, the SGE fusion weight $\beta$ is treated as a continuous action sampled from a Normal distribution $\mathcal{N}(\mu, \sigma^2)$ during training to encourage exploration.

**Reward and Policy Update.** To guide the learning of both APC and SGE, we define a sparse reward $R$ based on the VLLM's final decision accuracy. The reward is determined by comparing the extracted prediction (*e.g.*, "Yes/No" or "A/B/C") with the ground truth:

$$R = \begin{cases} +1.0 & \text{if prediction} = \text{ground truth,} \\ -1.0 & \text{otherwise.} \end{cases} \tag{5}$$

We employ the REINFORCE algorithm (Williams, 1992; Sutton et al., 1998) with a moving baseline $B$ to update the policy parameters. The baseline is updated via an exponential moving average ($B \leftarrow \varphi \cdot B + (1 - \varphi) \cdot R$) to reduce gradient variance. The objective is to minimize the following joint loss:

$$\mathcal{L} = -(\log P(\mathbf{a}_{apc} \mid \mathbf{H}, \mathbf{D}) + \log P(\mathbf{a}_{sge} \mid \mathbf{H})) \cdot (R - B) \tag{6}$$

where $\mathbf{a}_{apc}$ and $\mathbf{a}_{sge}$ represent the sampled perturbation mask and fusion weight, respectively. Further details are provided in Sec. C.5.

## 4. Experiments

### 4.1. Experimental Settings

**Models** We evaluate 39 VLLMs spanning 10 open-source and 3 proprietary model families, and include a human baseline for reference. Model details and implementation details are provided in Sec. D.1 and Sec. D.3, respectively.

**Evaluation Metrics.** We adopt Accuracy as the primary metric for both tasks, and additionally report task-specific diagnostic metrics, including Yes/No Accuracy for YNQA and macro-averaged Precision, Recall, and F1-score for MCQA. Detailed definitions are provided in Sec. D.2.

### 4.2. Benchmarking Models on OmniVCHall

Tab. 4 systematically reports the performance of 39 current VLLMs on OmniVCHall on Accuracy. The multi-dimensional analysis reveals the following:

**Obs.❶ Proprietary models lead in performance, though top-tier open-source models are narrowing the gap; a significant gap persists between AI and stable human performance.** As shown in Fig. 5, proprietary models such as Gemini-3-pro (0.77 avg) and GPT-5.2 (0.76 avg) establish a high performance floor with superior architectural robustness across diverse tasks. Within the open-source community, a stark performance divide exists where smaller models like VideoLLaMA3-2B (0.54 avg) struggle with spatiotemporal grounding, while high-end scaling in Qwen3-VL-Thinking-235B (0.76 avg) effectively achieves parity with proprietary state-of-the-art capabilities. Despite these

*Table 4.* Benchmark results on Accuracy. The shaded cells highlight the best and second-best results in the open-source and commercial models, respectively. The ***best model result*** among all models is highlighted. All results can be seen in Sec. D.4

| Model | Size | Real | Generated | Avg |
|---|---|---|---|---|
| Human | - | **0.95** | **0.94** | **0.95** |
| *Open-source Models* | | | | |
| VideoChat-Flash (Li et al., 2024c) | 2B | 0.55 | 0.57 | 0.56 |
| VideoLLaMA3 (Zhang et al., 2025) | 2B | 0.51 | 0.58 | 0.54 |
| Qwen3-VL-Instruct (Bai et al., 2025a) | 2B | 0.59 | 0.67 | 0.63 |
| Qwen3-VL-Thinking (Bai et al., 2025a) | 2B | 0.62 | 0.69 | 0.65 |
| Qwen2.5-VL-Instruct (Bai et al., 2025b) | 3B | 0.52 | 0.61 | 0.57 |
| InternVL3.5 (Wang et al., 2025) | 4B | 0.61 | 0.66 | 0.64 |
| MiniCPM-V-4 (Yu et al., 2025) | 4B | 0.57 | 0.63 | 0.60 |
| Qwen3-VL-Instruct (Bai et al., 2025a) | 4B | 0.64 | 0.72 | 0.68 |
| Qwen3-VL-Thinking (Bai et al., 2025a) | 4B | 0.67 | 0.73 | 0.70 |
| Molmo2 (Clark et al., 2026) | 4B | 0.63 | 0.68 | 0.66 |
| LLaVA-NeXT-Video (Zhang et al., 2024b) | 7B | 0.44 | 0.49 | 0.47 |
| VideoChat-Flash (Li et al., 2024c) | 7B | 0.58 | 0.60 | 0.59 |
| VideoLLaMA3 (Zhang et al., 2025) | 7B | 0.59 | 0.65 | 0.62 |
| Qwen2.5-VL-Instruct (Bai et al., 2025b) | 7B | 0.60 | 0.69 | 0.64 |
| InternVL3.5 (Wang et al., 2025) | 8B | 0.60 | 0.67 | 0.64 |
| MiniCPM-V-4.5 (Yu et al., 2025) | 8B | 0.64 | 0.68 | 0.66 |
| Qwen3-VL-Instruct (Bai et al., 2025a) | 8B | 0.65 | 0.74 | 0.70 |
| Qwen3-VL-Thinking (Bai et al., 2025a) | 8B | 0.67 | 0.74 | 0.71 |
| Molmo2 (Clark et al., 2026) | 8B | 0.65 | 0.69 | 0.67 |
| Kimi-VL-Instruct (Team et al., 2025) | 16B | 0.64 | 0.72 | 0.68 |
| Kimi-VL-Thinking (Team et al., 2025) | 16B | 0.68 | 0.73 | 0.71 |
| InternVL3.5 (Wang et al., 2025) | 30B | 0.62 | 0.67 | 0.65 |
| Qwen2.5-VL-Instruct (Bai et al., 2025b) | 32B | 0.62 | 0.71 | 0.67 |
| Qwen3-VL-Instruct (Bai et al., 2025a) | 32B | 0.69 | 0.76 | 0.72 |
| Qwen3-VL-Thinking (Bai et al., 2025a) | 32B | 0.71 | 0.77 | 0.74 |
| LLaVA-NeXT-Video(Zhang et al., 2024b) | 34B | 0.47 | 0.53 | 0.50 |
| GLM-4.5v (Hong et al., 2025) | 108B | 0.47 | 0.50 | 0.49 |
| GLM-4.6v-flash (Hong et al., 2025) | 108B | 0.44 | 0.49 | 0.46 |
| GLM-4.6v (Hong et al., 2025) | 108B | 0.49 | 0.52 | 0.51 |
| Qwen3-VL-Instruct (Bai et al., 2025a) | 235B | 0.71 | 0.77 | 0.74 |
| Qwen3-VL-Thinking (Bai et al., 2025a) | 235B | 0.73 | ***0.78*** | 0.76 |
| *Proprietary Models* | | | | |
| GPT-4o (OpenAI, a) | - | 0.70 | 0.73 | 0.71 |
| GPT-5 (OpenAI, b) | - | 0.72 | 0.75 | 0.73 |
| GPT-5.2 (OpenAI, c) | - | 0.74 | 0.77 | 0.76 |
| Gemini-2.5-flash (DeepMind, a) | - | 0.71 | 0.75 | 0.73 |
| Gemini-2.5-pro (DeepMind, b) | - | 0.71 | 0.74 | 0.73 |
| Gemini-3-pro (DeepMind, c) | - | ***0.76*** | 0.77 | ***0.77*** |
| Doubao-Seed-1.6 (Volcengine, a) | - | 0.71 | 0.75 | 0.73 |
| Doubao-Seed-1.8 (Volcengine, b) | - | 0.73 | 0.77 | 0.75 |

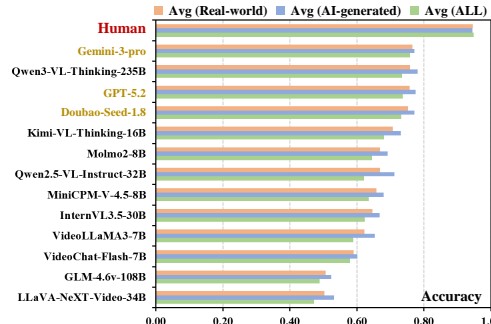

*Figure 5.* Comparative performance of the leading representatives from each model family. Human performance is highlighted at the top to establish a ceiling for evaluating current VLLM capabilities.

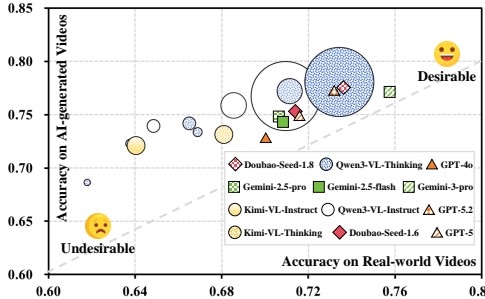

*Figure 6.* Performance correlation between different domains across VLLMs. For open-source models, circle sizes represent parameter scale: the Qwen3-VL series ranges from 2B to 235B, and the Kimi-VL series is 16B. The visualization further distinguishes between the Thinking and Instruct reasoning paradigms.

advancements, a substantial 18% accuracy gap remains between the best-performing AI (Gemini-3-pro, 0.77 avg) and the human baseline (0.95 avg), whose remarkably consistent performance across all complexities serves as critical evidence that OmniVCHall is a well-calibrated diagnostic tool, ensuring that degradation is strictly attributable to inherent reasoning failures rather than dataset bias.

**Obs.❷ Scale and reasoning paradigms influence performance; real-world videos are more challenging than generated ones—this gap motivated our targeted investigation into the intricate compositional hallucinations.** As shown in Fig. 6, models utilizing the "Thinking" paradigm tend to outperform their "Instruct" counterparts, such as

Qwen3-VL-Thinking-2B (0.65 avg) *vs.* Qwen3-VL-Instruct-2B (0.63 avg). suggesting that internal reasoning processes help mitigate logical biases across modalities. Larger models generally perform better. Proprietary models show consistent improvements over iterations, such as GPT-5 (0.73 avg) outperforming GPT-4o (0.71 avg). The positioning of most models above the diagonal in Fig. 6 highlights a significant domain bias: current VLLMs exhibit a higher risk of hallucination in real-world scenarios than in AI-generated content. While AI-generated content is curated to trigger hallucinations through unseen scenarios, our findings reveal that they still lack the complex dynamics and lighting of real-world videos. Crucially, this gap also reveals that the primary bottleneck is not scenario novelty but the dense spatiotemporal entanglement of real-world events, which triggers more elusive compositional hallucinations that are far more difficult to resolve. Based on this, we introduce OmniVCHall as a rigorous diagnostic tool and propose TriCD to calibrate the model's decoding boundaries against such multi-faceted logical contradictions.

**Obs.❸ Performance decreases with compositional complexity; camera-based reasoning is a dominant failure mode; adversarial options expose pervasive model in-**

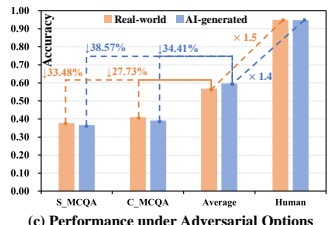

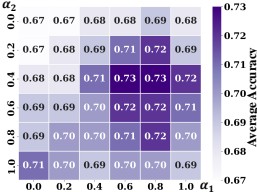

*Figure 7.* Performance analysis on OmniVCHall. (a) Accuracies decline from S_YNQA to C_MCQA. (b) Performance across eight types reveals that models struggle most with the novel camera type. (c) Significant accuracy drops under adversarial options reveal a severe lack of predictive stability in VLLMs.

*Figure 8.* Hyperparameter sensitivity. The optimal combination is $\alpha_1 = 0.8$ and $\alpha_2 = 0.4$.

**stability.** As shown in the left panel of Fig. 7, accuracy drops from S_YNQA to C_MCQA, highlighting the need for compositional hallucination detection. The middle panel in Fig. 7 indicates that models exhibit the highest error rates when addressing camera-based movements and perspective shifts. This validates our introduction of this new category, as it uncovers a systematic deficiency in perceiving "observer-perspective dynamics". The right panel shows that adversarial options significantly increase hallucination rates, pointing to higher instability under such conditions. These findings underline the challenges of OmniVCHall and establish a foundation for developing more robust models. Driven by these insights, we specifically engineered TriCD, incorporating APC to suppress the identified reasoning instabilities and SGE to resolve the pervasive spatiotemporal grounding deficiencies in complex scenarios.

**Summary.** OmniVCHall not only reveals a significant performance gap between current models and humans ($>0.15$) but also precisely localizes the weaknesses in spatiotemporal logic, complex reasoning, and camera dynamics.

### 4.3. Main Results of TriCD

We evaluate the effectiveness of TriCD by comparing it with representative contrastive decoding baselines, including MotionCD (Kong et al., 2025), TCD (Zhang et al., 2024a), and DINO-HEAL (Li et al., 2025a) (as detailed in Sec. D.5).

As shown in Tab. 5, TriCD consistently delivers substantial performance gains, averaging 10%, across diverse models. These results highlight its robust scalability and plug-and-play compatibility with various open-source VLLMs. Compared to existing baselines, our TriCD significantly outperforms methods that rely on uniform perturbations or a single saliency map. While MotionCD and TCD provide marginal gains ($\approx 2.93\%$) by targeting temporal inconsistencies. DINO-HEAL shows limited improvement ($\approx 2.86\%$) in dynamic scenarios. In contrast, our triple-pathway framework achieves a much higher performance ceiling by concurrently leveraging context-aware negative distraction and spatiotemporal evidence enhancement. We also report the inference efficiency in Sec. D.6. Another

*Table 5.* Main results on the test set. The shaded cells highlight the backbone's and our method's results. The final column reports the relative percentage against the original backbone.

| Model | S_YNQA | C_YNQA | S_MCQA | C_MCQA | Avg | - |
|---|---|---|---|---|---|---|
| Qwen3-VL-Instruct-8B | 0.75 | *0.69* | 0.61 | *0.53* | 0.67 | - |
| +MotionCD | 0.76 | *0.70* | 0.62 | *0.54* | 0.68 | ↑1.82% |
| +TCD | 0.77 | *0.72* | 0.62 | *0.55* | 0.69 | ↑3.61% |
| +DINO-HEAL | 0.77 | *0.72* | 0.62 | *0.54* | 0.68 | ↑1.99% |
| +TriCD | 0.79 | *0.78* | 0.66 | *0.64* | 0.73 | ↑9.61% |
| VideoLLaMA3-7B | 0.67 | *0.61* | 0.56 | *0.48* | 0.60 | - |
| +MotionCD | 0.68 | *0.62* | 0.57 | *0.48* | 0.61 | ↑2.34% |
| +TCD | 0.69 | *0.63* | 0.57 | *0.49* | 0.62 | ↑3.41% |
| +DINO-HEAL | 0.69 | *0.63* | 0.57 | *0.48* | 0.62 | ↑2.92% |
| +TriCD | 0.74 | *0.69* | 0.63 | *0.55* | 0.67 | ↑12.03% |
| LLaVA-NeXT-Video-7B | 0.54 | *0.49* | 0.38 | *0.33* | 0.45 | - |
| +MotionCD | 0.55 | *0.51* | 0.38 | *0.34* | 0.47 | ↑2.97% |
| +TCD | 0.56 | *0.52* | 0.39 | *0.34* | 0.47 | ↑4.47% |
| +DINO-HEAL | 0.54 | *0.51* | 0.39 | *0.35* | 0.47 | ↑3.22% |
| +TriCD | 0.57 | *0.57* | 0.41 | *0.42* | 0.51 | ↑11.48% |
| InternVL3.5-8B | 0.70 | *0.67* | 0.54 | *0.49* | 0.62 | - |
| +MotionCD | 0.71 | *0.69* | 0.55 | *0.50* | 0.63 | ↑1.65% |
| +TCD | 0.71 | *0.69* | 0.55 | *0.51* | 0.64 | ↑2.29% |
| +DINO-HEAL | 0.72 | *0.68* | 0.57 | *0.51* | 0.64 | ↑2.75% |
| +TriCD | 0.73 | *0.77* | 0.58 | *0.58* | 0.68 | ↑9.24% |
| VideoChat-Flash-7B | 0.60 | *0.57* | 0.54 | *0.50* | 0.56 | - |
| +MotionCD | 0.62 | *0.59* | 0.57 | *0.53* | 0.59 | ↑3.94% |
| +TCD | 0.62 | *0.58* | 0.56 | *0.52* | 0.58 | ↑2.84% |
| +DINO-HEAL | 0.62 | *0.59* | 0.56 | *0.52* | 0.58 | ↑3.44% |
| +TriCD | 0.70 | *0.66* | 0.58 | *0.54* | 0.62 | ↑9.76% |

critical observation is that our TriCD provides the most substantial improvements in compositional (C_) tasks. For instance, Qwen3-VL-Instruct-8B sees a 9% jump in C_YNQA and an 11% increase in C_MCQA. This performance leap is a direct result of the synergistic dual action of the APC and SGE modules, which together provide a robust "upper and lower bound" calibration for the model's predictions.

As shown in Tab. 6, TriCD is trained only on OmniVCHall and then directly evaluated on other benchmarks (Video-Hallucer (Wang et al., 2024b) and EventHallusion (Zhang et al., 2024a)). Despite this setting, TriCD still achieves consistent gains, averaging 11%, over the original backbone, outperforming other methods. These results suggest that the learned RL-based calibration policy transfers beyond the training benchmark and that OmniVCHall covers error patterns that generalize to other video benchmarks.

*Table 6.* Cross-benchmark transfer results. The shaded cells highlight the backbone's and our method's results. The final column reports the relative percentage against the original backbone.

| Model | VideoHallucer | EventHallusion | Avg | - |
|---|---|---|---|---|
| Qwen3-VL-Instruct-8B | 0.52 | 0.65 | 0.59 | - |
| +MotionCD | 0.53 | 0.64 | 0.58 | ↓0.60% |
| +TCD | 0.53 | 0.66 | 0.59 | ↑1.19% |
| +DINO-HEAL | 0.53 | 0.66 | 0.60 | ↑1.70% |
| +TriCD | 0.56 | 0.67 | 0.62 | ↑5.48% |
| VideoLLaMA3-7B | 0.39 | 0.67 | 0.53 | - |
| +MotionCD | 0.39 | 0.68 | 0.53 | ↑1.00% |
| +TCD | 0.41 | 0.66 | 0.53 | ↑1.68% |
| +DINO-HEAL | 0.39 | 0.67 | 0.53 | ↑1.04% |
| +TriCD | 0.48 | 0.69 | 0.58 | ↑11.02% |
| LLaVA-NeXT-Video-7B | 0.32 | 0.49 | 0.41 | - |
| +MotionCD | 0.32 | 0.50 | 0.41 | ↑1.11% |
| +TCD | 0.34 | 0.51 | 0.43 | ↑4.94% |
| +DINO-HEAL | 0.33 | 0.49 | 0.41 | ↑1.60% |
| +TriCD | 0.44 | 0.51 | 0.47 | ↑17.06% |
| InternVL3.5-8B | 0.50 | 0.56 | 0.53 | - |
| +MotionCD | 0.50 | 0.57 | 0.53 | ↑0.61% |
| +TCD | 0.50 | 0.57 | 0.54 | ↑1.09% |
| +DINO-HEAL | 0.50 | 0.57 | 0.54 | ↑0.92% |
| +TriCD | 0.54 | 0.58 | 0.56 | ↑5.59% |
| VideoChat-Flash-7B | 0.36 | 0.56 | 0.46 | - |
| +MotionCD | 0.37 | 0.57 | 0.47 | ↑1.34% |
| +TCD | 0.37 | 0.57 | 0.47 | ↑2.53% |
| +DINO-HEAL | 0.37 | 0.57 | 0.47 | ↑2.63% |
| +TriCD | 0.44 | 0.67 | 0.56 | ↑20.22% |

*Table 7.* Ablation study of TriCD. The shaded cells highlight the backbone's and our method's results. The final column reports the relative percentage against the original backbone.

| Model | S_YNQA | C_YNQA | S_MCQA | C_MCQA | Avg | - |
|---|---|---|---|---|---|---|
| Qwen3-VL-Instruct-8B | 0.75 | *0.69* | 0.61 | *0.53* | 0.67 | - |
| +TriCD | 0.79 | *0.78* | 0.66 | *0.64* | 0.73 ↑9.61% | |
| +TriCD w/o APC | 0.78 | *0.72* | 0.62 | *0.54* | 0.69 ↑2.54% | |
| +TriCD w/o SGE | 0.78 | *0.76* | 0.63 | *0.61* | 0.71 ↑6.72% | |
| +TriCD w/ RP | 0.70 | *0.69* | 0.54 | *0.52* | 0.63 ↓5.87% | |
| +TriCD w/ STE | 0.76 | *0.75* | 0.63 | *0.62* | 0.69 ↓3.89% | |
| +TriCD w/o DINO | 0.78 | *0.77* | 0.64 | *0.63* | 0.72 ↑7.74% | |
| +TriCD w/o Motion | 0.79 | *0.77* | 0.64 | *0.62* | 0.72 ↑7.65% | |

## 4.4. Ablation Study

As shown in Tab. 7, we conduct ablation studies. Removing APC leads to a more pronounced performance drop (0.73 → 0.69) compared to ablating SGE (0.73 → 0.71). This degradation is especially evident in compositional hallucination tasks: without APC, accuracy plummets by 0.06 on C_YNQA and by 0.10 on C_MCQA. This suggests that adaptive negative suppression plays a more dominant role in resolving hallucinations. Using random perturbation (RP) results in a collapse to 0.63, worse than the backbone (0.67), proving that APC's optimized selection is vital to avoid destructive interference during decoding. Straight-through estimator (STE) relies on a biased surrogate gradient and optimizes a relaxed proxy rather than the actual hard routing used at test time (Bengio et al., 2013; Jang et al., 2016).

Within the SGE module, ablating DINOv3 spatial or motion-based temporal features reduces performance to 0.72. The

specific drop in C_MCQA without motion cues (0.64 → 0.62) proves the necessity of inter-frame dynamics for complex temporal reasoning.

Finally, sensitivity analysis in Fig. 8 confirms that the optimal balance is achieved at $\alpha_1 = 0.8$ and $\alpha_2 = 0.4$ (as detailed in Sec. D.7). This peak validates that precisely calibrating the relative strengths of negative distraction and positive evidence is critical for robust spatiotemporal grounding.

## 5. Conclusion

In this paper, we introduce the first comprehensive benchmark (**OmniVCHall**) for evaluating both isolated and compositional hallucinations in video multimodal large language models. We also develop a triple-pathway contrastive decoding framework (**TriCD**) that unifies adaptive negative suppression with saliency-guided enhancement for mitigating *compositional* hallucinations. We hope this work will inspire future research on principled modeling and mitigation of compositional hallucinations, towards more robust spatiotemporal reasoning and grounding mechanisms.

## Acknowledgement

This work is supported by the National Natural Science Foundation of China (Grant No. 62502317), and the Guangdong Basic and Applied Basic Research Foundation (Grant No. 2026A1515011198).

## Impact Statement

This paper introduces the OmniVCHall benchmark and the TriCD framework to diagnose and mitigate compositional hallucinations in video multimodal large language models. By improving spatiotemporal grounding, our work helps reduce misleading video understanding outputs in applications such as video analysis and human-AI interaction. However, OmniVCHall is built from carefully selected and human-verified videos, and does not fully cover noisier, longer, or weakly observable real-world scenarios. Therefore, the reported gains should not be interpreted as robustness guarantees for unconstrained or safety-critical deployment. Evaluating such settings remains important future work.

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

# Supplementary Material for:

## *Learning to Decode Against Compositional Hallucination in Video Multimodal Large Language Models*

## Appendix Overview

# A. Related Work

## A.1. Video Multimodal Large Language Models

Multimodal large language models have achieved remarkable success in image understanding (Liu et al., 2023; Li et al., 2026a; Achiam et al., 2023; Li et al., 2026b; Fang et al., 2023; Zheng et al., 2022; Fang et al., 2026; Zheng et al., 2023; Li et al., 2025b; Tan et al., 2025; Zhao et al., 2025) by effectively aligning visual features with linguistic representations in a shared latent space (Radford et al., 2021; Li et al., 2025e; Diao et al., 2026; Li et al., 2023b; Jiang et al., 2025; Li et al., 2025d). Building on this foundation, the field has increasingly pivoted toward video multimodal large language models (VLLMs) to model complex temporal dynamics across high-dimensional visual sequences (Zou et al., 2024; Tang et al., 2025). Most recently, frontier closed source models such as GPT-5.2 (OpenAI, c) and Gemini-3-Pro (DeepMind, c) have emerged, as well as high-performance open source models including GLM-4.6V (Hong et al., 2025), Qwen3-VL (Bai et al., 2025a), InternVL3.5 (Wang et al., 2025), and VideoLLaMA3 (Zhang et al., 2025). By incorporating sophisticated spatiotemporal compression techniques such as token reduction (Li et al., 2024d), merging (Jin et al., 2024), or pooling (Xu et al., 2024), VLLMs have achieved unprecedented performance in perception and efficiency. Early video reasoning methods focused on learning aligned spatiotemporal representations for basic temporal understanding (Yu et al., 2023). More recent work has shifted toward multi-step video reasoning, enabling iterative inference over longer temporal contexts (Luo et al., 2026; Pu et al., 2025; Liao et al., 2025). However, current VLLMs remain profoundly susceptible to severe hallucination issues, which continue to pose a major obstacle for their deployment in reliability-critical scenarios (Rawte et al., 2023; Sahoo et al., 2024).

## A.2. Video Hallucination Benchmark

The rapid evolution of VLLMs has catalyzed the development of various evaluation benchmarks such as SEED-Bench (Li et al., 2023a; 2024a), MVBench (Li et al., 2024b), and VideoMME (Fu et al., 2025). And the quantification of visual hallucinations has become a pivotal necessity for ensuring reliability. Consequently, specialized benchmarks including VidHal (Choong et al., 2024), VideoCon (Bansal et al., 2024), VideoHallucer (Wang et al., 2024b), EventHallusion (Zhang et al., 2024a), MHBench (Kong et al., 2025), VidHalluc (Li et al., 2025a), and ARGUS (Rawal et al., 2025) have been introduced to detect specific visual inconsistencies. Despite these advancements, the evaluative scope of existing benchmarks remains notably fragmented. Most current efforts focus on isolated hallucination types such as action-based (Kong et al., 2025) or temporal-based (Choong et al., 2024; Li et al., 2025a) while neglecting essential video-centric elements, particularly camera-lens dynamics. Furthermore, these works often overlook the complexity of compositional hallucinations (Yang et al., 2026; Chytas et al., 2025; Hu et al., 2025; He et al., 2026), where multiple hallucination types concurrently manifest within a single query. For image understanding, compositional hallucinations have been actively explored, spanning from the discovery of attribute binding failures (Yang et al., 2026) and multi-dimensional evaluations (VS et al., 2026; Lim et al., 2025), to targeted mitigations like reminder compositions (Chytas et al., 2025) and LLM-guided extraction (Li et al., 2025c). However, directly extending these static paradigms to the video domain introduces profound temporal complexities. In dynamic scenes, compositional errors frequently trigger cascading reasoning failures across time, rather than remaining as isolated static mistakes. These limitations underscore the need for a more comprehensive and integrated evaluation protocol. As summarized in Tab. 2, our OmniVCHall fills this gap by introducing eight fine-grained hallucination types and complex multi-type testing to assess the robustness of VLLMs across both real-world and generated videos.

## A.3. Hallucination Mitigation Approach

Hallucination manifests as generated responses that contain information inconsistent with the actual visual content, typically arising from modal misalignment, imbalanced training data, or over-reliance on prior linguistic knowledge (Bansal et al., 2024; Bai et al., 2024). In complex multi-step tasks, these multimodal hallucinations are often exacerbated by phenomena like "visual thinking drift" (Luo et al., 2026) and intermediate "mirages" (Dong et al., 2026). Various strategies have been proposed to mitigate these inconsistencies. For instance, a contrast-caption fine-tuning strategy (Bansal et al., 2024) improves video-language correspondence. The Self-PEP framework (Wang et al., 2024b) utilizes a Predict-Explain-Predict loop to verify and revise initial judgments through generated explanations. DINO-HEAL (Li et al., 2025a) addresses the vulnerability of VLLMs to semantically similar distractors, attributing such hallucinations to the inherent inductive bias of CLIP-series encoders (Radford et al., 2021; Zhai et al., 2023). Furthermore, a method employing improved positional encoding and Direct Preference Optimization (Lu et al., 2025) targets semantic aggregation in long-video understanding. Additionally, a hierarchical multimodal consistency approach reduces hallucinations by enforcing structural alignment across

multiple granularities of video and language through specialized training objectives (Dang et al., 2025). And, techniques such as dynamically injecting contextual reminders during inference (Chytas et al., 2025) and utilizing external language models to explicitly parse visual structures (Li et al., 2025c) have been proposed to prevent the semantic unbinding of objects and attributes.

Among these advancements, Contrastive Decoding (CD) has emerged as a powerful training-free alternative that refines token predictions by adjusting logit streams during inference (Leng et al., 2024; Park et al., 2025). A temporal CD approach (Zhang et al., 2024a) has been developed by contrasting original frames with temporally distorted sequences. Similarly, MotionCD (Kong et al., 2025) penalizes logits generated from motion-corrupted negative samples. However, these methods are often restricted to predefined hallucination types and employ static, uniform perturbations, which lack the flexibility to adapt to diverse video content. To bridge these gaps, our TriCD framework unifies these two paradigms. It dynamically selects perturbation tools according to video and query context for adaptive negative suppression, while fusing spatial and temporal saliency for visual enhancement.

## B. OmniVCHall Benchmark Details

### B.1. Definition Details

**Minimal Evidence-Type Set.** The evidence-type set $G(V, T, A)$ describes the semantic conditions required by the query, rather than the number of errors made by a model. Each element in $G$ corresponds to one hallucination type in our taxonomy, such as object, attribute, action, relation, temporal order, or camera dynamics. For example, answering "What color is the car?" mainly requires an attribute-level judgment, leading to $|G| = 1$. In contrast, answering "Is the red car turning left?" requires jointly verifying the object, its attribute, and its action, leading to $|G| \geq 2$.

This distinction is important because compositional hallucination is defined by the structure of the required evidence, not merely by the co-occurrence of multiple wrong predictions. In a compositional case, the model may hallucinate one required element, multiple required elements, or correctly recognize individual elements but fail to bind them into a valid answer. All these cases are treated as compositional hallucinations because the answer depends on a jointly sufficient set of interdependent visual conditions. In addition, hallucination and perception/reasoning errors describe different aspects of model behavior. Hallucination targets the observed output-evidence inconsistency (the resulting outcome), whereas perception and reasoning are its underlying causes (Dong et al., 2026; Shao et al., 2025).

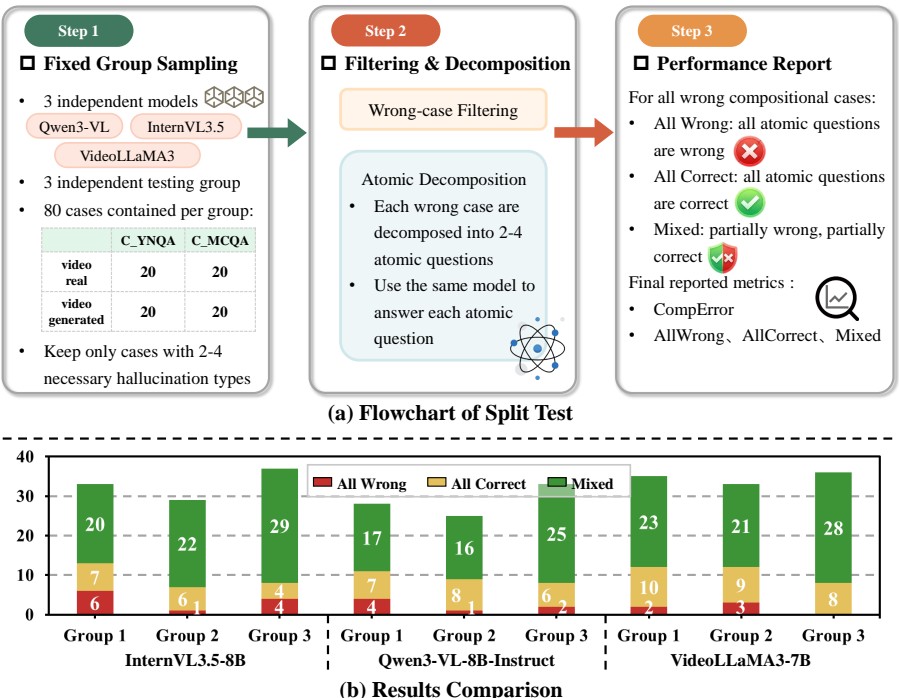

*Figure 9.* Atomic decomposition test.

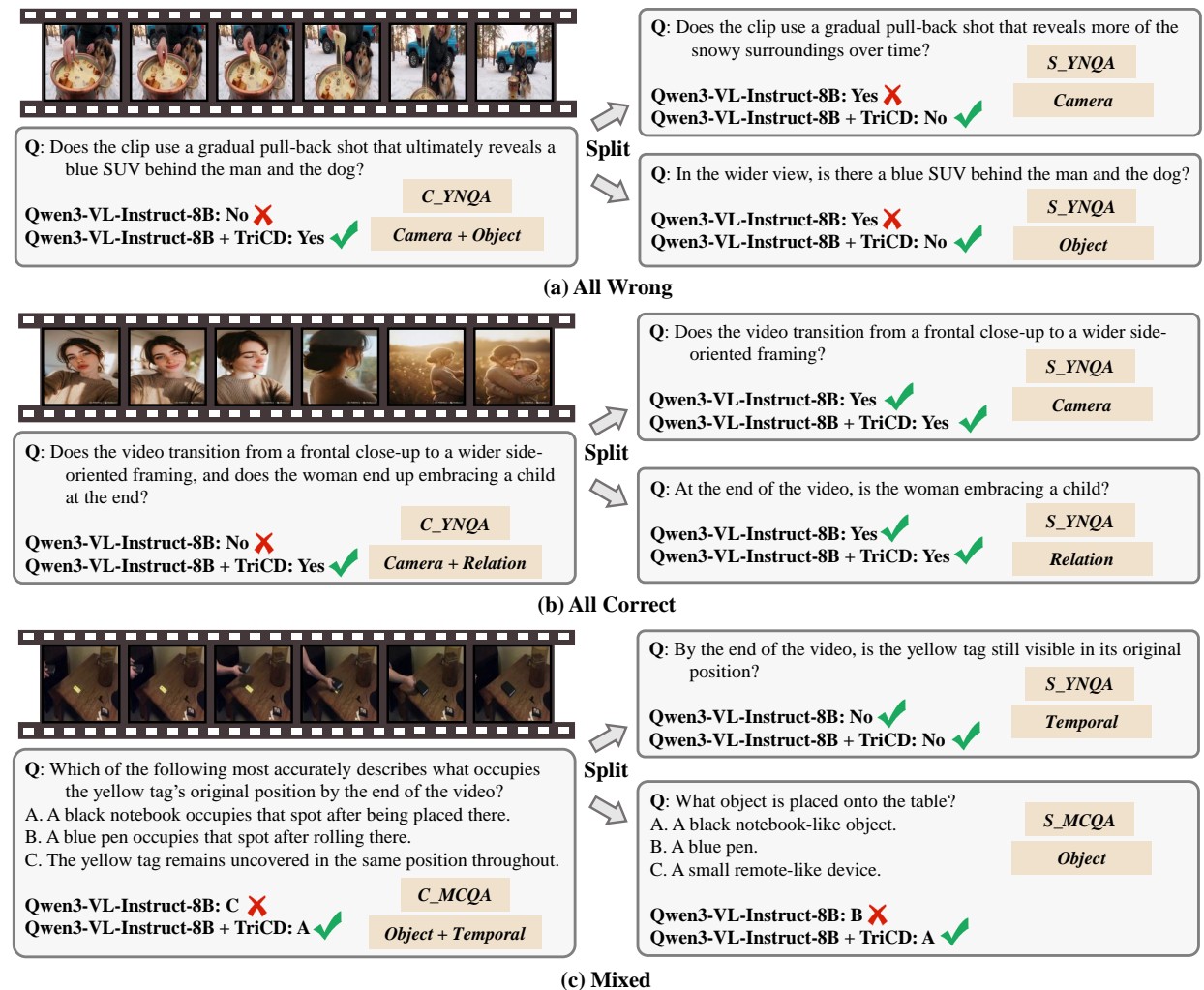

*Figure 10.* Illustrative examples of the three regimes in the atomic decomposition test.

**Atomic Decomposition Test.** To further analyze compositional failures, we decompose each wrong compositional case into its constituent single-type atomic questions. As shown in Fig. 9 (a), this decomposition allows us to distinguish whether the failure mainly comes from isolated perception errors, composition-level reasoning errors, or their interaction. We identify three representative patterns:

- **All Wrong**: all atomic questions fail, indicating that the compositional failure is accompanied by failures on every required evidence type.

- **All Correct**: all atomic questions are answered correctly, but the compositional question fails, indicating a composition-dominant reasoning error.

- **Mixed**: only part of the atomic questions fail, indicating coupled perception and reasoning failures.

As shown in Fig. 9 (b), the "All Wrong" (8.0%) and "Mixed" (69.4%) patterns demonstrate that 77.4% of compositional failures are driven exactly by the co-occurrence of multiple hallucinations. The remaining "All Correct" (22.6%) cases represent pure compositional reasoning failures, where individual elements are perceived correctly, but their combination is not. This confirms that compositional hallucination is not simply equivalent to multiple simultaneous perception errors; it also includes cases where individual visual facts are recognized, but their binding or joint reasoning is incorrect.

**Re-annotation Audit.** We further conduct a blinded re-annotation audit to verify whether the proposed definition can be applied consistently. As shown in Fig. 11 (a), for each of three independently sampled groups, annotators receive 80 mixed cases covering S_YNQA, C_YNQA, S_MCQA, and C_MCQA, and only see the video, question, and answer. Their task is to recover the minimal necessary evidence-type set. We evaluate both the binary distinction between single and compositional cases and the recovery of the minimal evidence-type set. As shown in Fig. 11 (b), the high binary accuracy and Jaccard similarity indicate that the definition is reproducible and provides a stable criterion for identifying compositional hallucination cases.

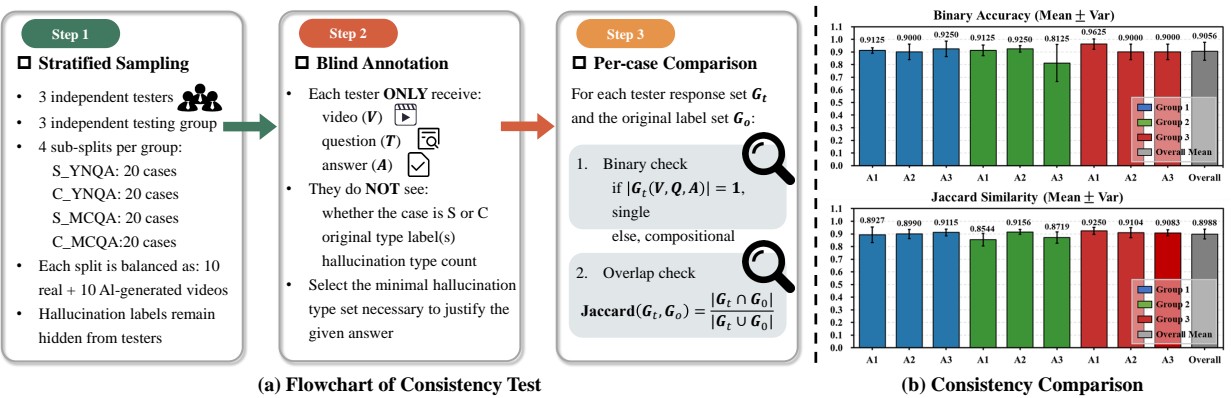

**(a) Flowchart of Consistency Test**  **(b) Consistency Comparison**

*Figure 11.* Blinded re-annotation audit for validating the single- vs. compositional- definition.

## B.2. Hallucination Type

Tab. 8 presents a formal taxonomy of the eight hallucination types featured in OmniVCHall, providing precise definitions and representative examples for each. This systematic categorization spans fundamental object properties and complex spatiotemporal dynamics, including our novel camera-centric type to isolate lens effects from physical motion. By establishing clear ground-truth boundaries, this table serves as a diagnostic reference for identifying specific perceptual and reasoning failures in VLLMs.

*Table 8.* Definition and examples of the eight hallucination types in OmniVCHall.

| Type | Definition | Example |
|------|------------|---------|
| Object | Presence or identity of physical entities. | Describing a "cat" when only a "dog" exists. |
| Scene | The overarching environment or setting. | Misidentifying a "hospital" as a "school". |
| Event | High-level semantic units or causal occurrences. | Misinterpreting a "goal scored" as a "missed shot". |
| Action | Physical movement or behavioral patterns. | Claiming a person is "running" while they are "walking". |
| Relation | Spatial or logical interactions between entities. | Stating a bowl is "on the table" when it is "under" it. |
| Attribute | Static properties like color, size, or material. | Describing a "red ball" as "blue". |
| Temporal | Chronological order or duration of occurrences. | Reversing the order of "pouring water" and "drinking". |
| Camera | Perception of cinematic lens dynamics. | Interpreting a "zoom-in" as an "object moving closer". |

*Table 9.* Statistics of video data sources in OmniVCHall.

| Type | Source | Count | Platform |
|------|--------|-------|----------|
| Real-world | VidHalluc (Li et al., 2025a) | 51 | https://huggingface.co/datasets/chaoyuli/VidHalluc |
| | Online Search | 100 | https://www.youtube.com/ |
| | MHBench (Kong et al., 2025) | 120 | https://drive.google.com/drive/folders/1INrzOafJe6uKFp0IZp1z-pdw9bq_YYpJ |
| | EventHallusion (Zhang et al., 2024a) | 152 | https://drive.google.com/file/d/1IPmx6Y80UrXwVPmZJh6zjCPHtlsw4p9n/view |
| AI-generated | Sora (OpenAI, d) | 34 | https://openai.com/sora |
| | SeedanceVideo (ByteDance) | 36 | https://www.seedance.ai |
| | Veo (Google) | 55 | https://deepmind.google/models/veo/ |
| | HailuoVideo (Minimax) | 55 | https://hailuoai.video/ |
| | HunyuanVideo (Tencent) | 70 | https://www.hunyuanvideo.org/ |
| | KlingVideo (Kuaishou) | 70 | https://app.klingai.com/global/ |
| | Wan (Alibaba) | 80 | https://wan.video/ |

## B.3. Data Source

Tab. 9 details the distribution of the 823 video clips across 11 heterogeneous sources. The dataset is balanced between established real-world benchmarks and synthetic content from frontier generative models like Sora (OpenAI, d), Veo (Google), and Wan (Alibaba). This multi-domain composition ensures that OmniVCHall provides a robust evaluation environment covering both authentic real-world dynamics and complex AI-generated artifacts.

## B.4. Prompt for QA Generation

Fig. 12 illustrates the structured prompt utilized for the automated generation of QA pairs. By providing multi-modal inputs, including the video and its corresponding ground-truth caption, the prompt employs dynamic slots to toggle between different question formats and complexity levels. This systematic approach ensures that the generated 9,027 QA pairs are accurately aligned with our fine-grained hallucination taxonomy while maintaining rigorous logical consistency across both isolated and compositional scenarios.

---

### System Prompt
You are an expert hallucination detection assistant for video-language models. Your task is to analyze the provided *{Video}* and *{Ground-truth Caption}* to generate challenging QA pairs that trigger specific spatiotemporal hallucinations in Video Large Language Models. These hallucinations may mislead or hinder visual-language reasoning. Each query may trigger one or multiple hallucination types. Below are the definitions and detailed analysis guidelines for each hallucination type.

### Hallucination Type Definitions
1. **Action**: Misunderstandings about actions occurring in the video.
2. **Scene**: Misunderstandings about the video's scene or background.
3. **Object**: Misunderstandings about the existence of objects in the video.
4. **Relation**: Misunderstandings about relationships between objects or actions.
5. **Attribute**: Misunderstandings about properties of objects or actions.
6. **Event**: Misunderstandings about whether events occurred or their specific content.
7. **Temporal**: Misunderstandings about the sequence or timing of events.
8. **Camera**: Misunderstandings about camera lens movement.

### Detailed Analysis Guidelines
1. **Action**: Queries about what someone or something is doing; verbs indicating movements or behaviors.
2. **Scene**: Queries about location, setting, weather, or background environment.
3. **Object**: Queries about presence, absence, or recognition of items, people, or entities.
4. **Relation**: Queries about spatial, physical, or semantic relations between entities or actions (e.g., "next to", "holding", "with").
5. **Attribute**: Queries about qualities such as color, shape, texture, size, number, or material.
6. **Event**: Queries about overall happenings or incidents (e.g., "What happens?", "Did something occur?").
7. **Temporal**: Queries involving order, timing, duration, "before/after", or "first/then" relations.
8. **Camera**: Queries involving camera movement, position, focus, etc.

### Task-Specific Constraints:
-mode: **[Yes/No Question-Answering]** or **[Multi-choice Question-Answering]**
-Complexity: **[Single Hallucination Type]** or **[compositional Hallucination Type]**

Here are some examples: **[QA Examples]**

Please return the content strictly according to the following format:
{
    "type": ,
    "question": ,
    **["option": ,]**
    "answer": ,
    "reasoning": ,
}

*Figure 12.* Prompt for Gemini-2.5-Pro (DeepMind, b) to generate QAs. *Terms* (*e.g.*, {Video} and {Ground-truth Caption}) represent the multi-modal input variables provided to the model. **Terms** (*e.g.*, **[Yes/No Question-Answering]** or **[Single Hallucination Type]**) denote dynamic slots that are selectively toggled based on the target question format (YNQA *vs.* MCQA) and complexity level (Simple *vs.* Compositional) required for the four evaluation sub-categories.

## B.5. Adversarial Option

Tab. 3 summarizes the distribution of adversarial option injections across different video types and question complexities in OmniVCHall. To enhance evaluation rigor, we selectively replaced standard distractors with adversarial options (*e.g.*, "All others are correct" and "All others are wrong"), maintaining a significant ratio (up to 33.84% in complex real-world scenarios) to challenge the models' predictive stability, as illustrated in Fig. 13 and Fig. 14. This injection strategy forces the model to verify every candidate option against the visual evidence rather than relying on a process of elimination or linguistic bias.

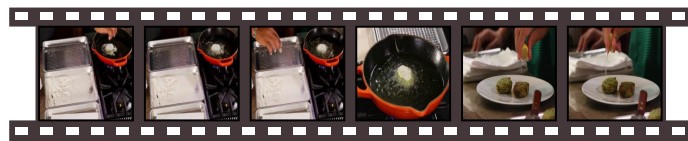

*Figure 13.* Illustration of the "All others are wrong" adversarial injection. (Left) The original MCQA includes a standard set of distractors and a grounded correct answer. (Right) The grounded answer is replaced with the adversarial option "All others are wrong", requiring the model to explicitly negate all other incorrect descriptions based on the visual evidence.

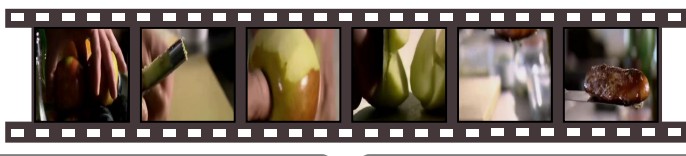

*Figure 14.* Illustration of the "All others are correct" adversarial injection. (Left) A standard MCQA with one correct action and two distractors. (Right) A modified QA where the original error option C is corrected to match the visual content, and option B is subsequently replaced with the adversarial option "All others are correct", testing the model's ability to recognize multiple valid descriptions simultaneously.

## B.6. QA Distribution

The detailed distribution of queries across specific hallucination types is provided in Tab. 10 and Tab. 11. Across both real-world and AI-generated domains, action emerges as the most frequently targeted hallucination type, while all eight categories—including the newly introduced camera type—are extensively covered across simple and complex QA formats.

*Table 10.* Distribution of QA pairs across different hallucination types in real-world scenarios.

| Type | Object | Scene | Event | Action | Relation | Attribute | Temporal | Camera |
|---|---|---|---|---|---|---|---|---|
| S_YNQA | 160 | 99 | 148 | 360 | 206 | 149 | 140 | 153 |
| C_YNQA | 467 | 294 | 464 | 525 | 414 | 470 | 306 | 202 |
| S_MCQA | 138 | 117 | 93 | 389 | 117 | 82 | 72 | 126 |
| C_MCQA | 378 | 260 | 256 | 434 | 267 | 440 | 106 | 91 |

*Table 11.* Distribution of QA pairs across different hallucination types in AI-generated scenarios.

| Type | Object | Scene | Event | Action | Relation | Attribute | Temporal | Camera |
|------|--------|-------|-------|--------|----------|-----------|----------|--------|
| S_YNQA | 254 | 187 | 174 | 330 | 272 | 224 | 199 | 173 |
| C_YNQA | 414 | 453 | 422 | 480 | 413 | 553 | 265 | 155 |
| S_MCQA | 123 | 151 | 62 | 284 | 146 | 83 | 91 | 172 |
| C_MCQA | 330 | 390 | 336 | 399 | 330 | 458 | 246 | 160 |

## B.7. Word Cloud

As shown in Fig. 15, the word cloud highlights the core focus of our benchmark, particularly emphasizing camera and temporal dynamics.

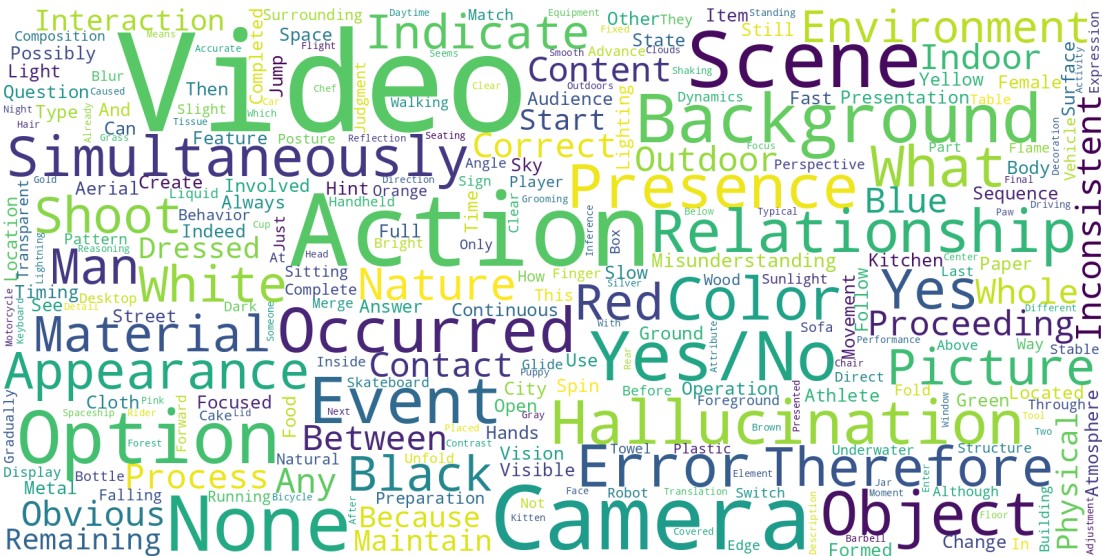

*Figure 15.* Illustration of a semantic word cloud.

## B.8. Inter-Annotator Agreement (IAA)

To ensure the reliability of the OmniVCHall ground truth, we conducted an IAA analysis across three critical stages:

- To validate the quality of the initial video descriptions, we randomly selected 15% of the video corpus for dual-annotation. Two independent annotators were tasked with identifying primary entities, key actions, and camera movements for each clip. We evaluated their agreement based on the presence and identity of these core spatiotemporal elements. This stage achieved a Cohen's Kappa of 0.84, representing "almost perfect" agreement according to standard Landis-Koch benchmarks (Landis & Koch, 1977).

- The automated QA generation process utilized Gemini-2.5-Pro (DeepMind, b). To assess the validity, two independent experts reviewed a stratified random sample of 1,354 pairs (15%). Each pair was evaluated against three criteria: (1) visual groundedness, (2) logical consistency of the options, and (3) accurate assignment, all with expertise in computer vision, yielding a Cohen's Kappa of 0.81 for the "Valid/Invalid" classification (Landis & Koch, 1977).

- The final reliability of the OmniVCHall ground truth is established through a rigorous human baseline evaluation. We recruited three independent annotators, all with expertise in computer vision, to complete the full evaluation suite. For each of the four evaluation sub-tasks (S_YNQA, C_YNQA, S_MCQA, C_MCQA), we observed that human accuracy remained remarkably consistent, with performance narrowly constrained between 0.91 and 0.96 regardless of the video source or question complexity. This low variance across sub-tasks demonstrates that the benchmark presents a uniform and fair difficulty level from a human cognitive perspective.

*Table 12.* Functional analysis of eight video perturbation tools.

| Type | Tool | Description | Explanation |
|---|---|---|---|
| Temporal | Sample | Performs uniform downsampling at a specific ratio to compress and omit intermediate continuous dynamic transitions. | Disrupts action continuity, forcing the model to reason with sparse temporal evidence. |
| | Reverse | Fully reverses the frame sequence chronologically, inverting the causal order while preserving individual frame content. | Penalizes models that rely on static cues rather than true temporal progression (*e.g.*, misidentifying "sitting" vs. "standing"). |
| | Shuffle | Randomly permutes the frame order to thoroughly destroy local temporal consistency. | Breaks the logical event chain, exposing models that over-rely on global scene context rather than sequential coherence. |
| Spatial | Blur | Applies a Gaussian kernel to systematically weaken high-frequency textures, edges, and fine-grained local details. | Challenges the model's ability to recognize object categories and existence when fine-grained visual evidence is degraded. |
| | Noise | Injects stochastic Gaussian noise to perturb pixel-level consistency while maintaining the global structure. | Exposes vulnerabilities in identifying textures, materials, and surface states that rely on pixel-level statistical accuracy. |
| | Grayscale | Strips all chromatic information by converting RGB frames into a unified luminance-based grayscale format. | Exposes reliance on color-based shortcuts for judging scene attributes, time of day, or specific material properties. |
| | Horizontal Mirror | Flips frames horizontally, preserving object identities while altering lateral spatial relationships. | Targets errors in judging camera panning and directional spatial relations (*e.g.*, "left-to-right" vs. "right-to-left" motion). |
| | Vertical Mirror | Flips frames vertically, inverting top-bottom relationships while maintaining visual consistency. | Penalizes misinterpretations of camera angles (*e.g.*, high vs. low-angle shots) and gravity-based spatial priors (*e.g.*, "above" vs. "below"). |

# C. TriCD Framework Details

## C.1. Tool Analysis

For each tool in $\mathcal{P}$, we construct the comprehensive functional analysis, as summarized in Tab. 12. These descriptions are formulated using a unified template as presented below:

*"This {Type} perturbation, specifically {Description}, which effectively {Explain}".*

## C.2. Visual Examples

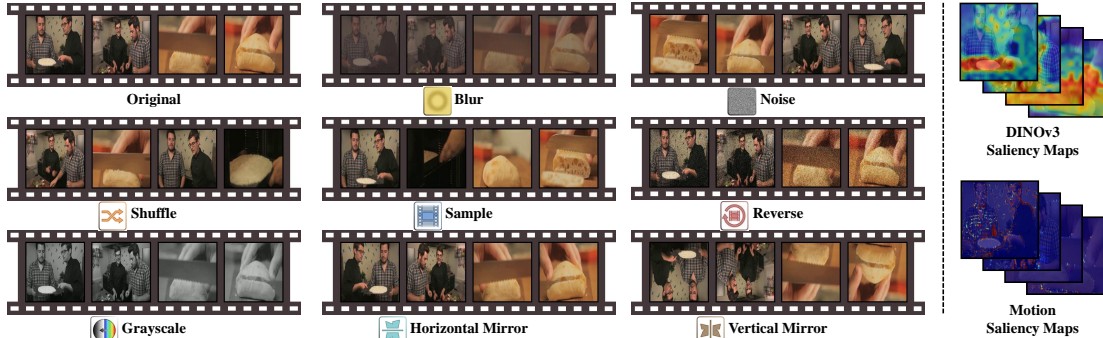

*Figure 16.* Visual illustration of the eight video perturbation tools (left) and the two saliency maps (right). To demonstrate the diverse effects of our perturbation, we extract representative frames from the same timestamp across the original and perturbed video sequences. These tools encompass both temporal disruptions (Reverse, Shuffle, Sample) and spatial/frame-level distortions (Blur, Noise, Grayscale, Mirror). The right panel showcases the DINOv3-based spatial saliency maps and optical flow-based motion saliency maps.

## C.3. Details of Adaptive Perturbation Controller

Alg. 1 defines the execution logic of the Adaptive Perturbation Controller, which transforms the raw video into a semantically-targeted negative sample. To facilitate precise tool selection, we introduce two specialized learnable embeddings:

- Type Embeddings ($\mathbf{E}_{type}$): These embeddings are added to the projected hidden states of both the video-query context ($\mathbf{E}_{type}^{vq}$) and the tool descriptions ($\mathbf{E}_{type}^{tool}$). This allows the cross-attention mechanism to explicitly distinguish between the semantic "request" (context) and the "resource" (available tools), improving the alignment accuracy.

- ID Embeddings ($\mathbf{E}_{id}$): Each of the eight perturbation tools is assigned a unique learnable ID embedding. These are injected into the tool description features to ensure that even if two tools have semantically similar descriptions, the APC can maintain distinct routing preferences and ranking scores for each.

---

**Algorithm 1** Algorithm workflow of APC

---

**Require:** Raw Video $V$; Query $T$; Hidden States $\mathbf{H}$; Tool Description Embeddings $\{\mathbf{D}_i\}_{i=1}^8$; Threshold $\gamma$.
**Ensure:** Perturbed Video $V^-$; Negative Logit $q_t^n$.
1: $\mathbf{H}' \leftarrow \text{Proj}(\mathbf{H}) + \mathbf{E}_{type}^{vq}$    // Apply projection & type embedding
2: **for** each tool $i \in \{1, \ldots, 8\}$ **do**
3:    $\mathbf{D}_i' \leftarrow \text{Proj}(\mathbf{D}_i) + \mathbf{E}_{type}^{tool} + \mathbf{E}_{id}^i$    // Incorporate type & ID embeddings
4: **end for**
5: $\mathbf{Q}_{cond} \leftarrow \text{Cross-Attn}(\mathbf{Q}, \mathbf{H}')$    // Stage A: Contextual conditioning
6: **for** each tool $i \in \{1, \ldots, 8\}$ **do**
7:    $\mathbf{Q}_{route}^i \leftarrow \text{Cross-Attn}(\mathbf{Q}_{cond}, \mathbf{D}_i')$    // Stage B: Tool-specific routing
8:    $\mathbf{r}_i \leftarrow \text{Mean}(\mathbf{Q}_{route}^i)$    // Token aggregation
9:    $s_i \leftarrow \text{ScoringHead}(\mathbf{r}_i)$    // Generate logit
10: **end for**
11: $\mathbf{P} \leftarrow \text{Softmax}(\mathbf{s})$    // Probability distribution over tools
12: $\mathcal{I}_{sorted} \leftarrow \text{Argsort}(\mathbf{P}, \text{descending})$    // Rank by relevance
13: Find $k$ such that $\sum_{j=1}^k \mathbf{P}_j \geq \gamma$    // Cumulative thresholding
14: $\mathcal{T}_{select} \leftarrow \{\tau_i \mid i \in \mathcal{I}_{sorted}[1:k]\}$
15: $V^- \leftarrow \mathcal{T}_{select}(V)$    // Apply selected perturbation
16: $q_t^n \leftarrow \text{logit}_\theta(y_t \mid V^-, T, y_{<t})$    // Obtain negative logit

---

## C.4. Details of Saliency-Guided Enhancement

**Spatial Saliency Extraction.** We utilize DINOv3 (Siméoni et al., 2025) as a spatial saliency extractor to capture foreground object importance. In DINOv3, the token sequence for each frame is structured as $\mathbf{X}_{DINO} = [\mathbf{x}_{cls}; \mathbf{x}_{reg}^{1\ldots4}; \mathbf{x}_p^{5\ldots5+N}]$, where $\mathbf{x}_{cls}$ is the [CLS] token, $\mathbf{x}_{reg}$ are 4 register tokens, and $\mathbf{x}_p$ are $N$ patch tokens. Following the self-attention mechanism of the Vision Transformer (Vaswani et al., 2017; Dosovitskiy et al., 2021), we extract the attention weights from the last transformer layer. Specifically, we compute the attention from the [CLS] token to all patch tokens to identify region-wise significance. Let $\mathbf{A}^{(h)} \in \mathbb{R}^{(N+5)\times(N+5)}$ be the attention matrix for head $h_D$, the unified spatial saliency $\mathbf{s}_{spa}$ is obtained by averaging across all $H_D$ heads:

$$\mathbf{s}_{spa} = \text{softmax}\left(\frac{1}{H_D} \sum_{h_D=1}^{H_D} \mathbf{A}_{[0,5:N+5]}^{(h_D)}\right). \tag{7}$$

**Temporal Saliency Extraction.** To capture dynamic transitions, the motion saliency extractor estimates the optical flow field $\mathbf{F} \in \mathbb{R}^{H \times W \times 2}$ between consecutive frames. We employ the Farneback algorithm (Farnebäck, 2003) to derive the raw motion magnitude $\mathbf{M}$ at each pixel:

$$\mathbf{M} = \sqrt{F_x^2 + F_y^2}, \tag{8}$$

where $F_x$ and $F_y$ represent the horizontal and vertical displacement components of the flow field $\mathbf{F}$, respectively. While $\mathbf{M}$ captures all pixel-level changes, it is inherently susceptible to background noise and global camera jitter. To refine this into a meaningful saliency signal, we apply a dual-stage filtering process. First, a spatial Gaussian filter is used to smooth the magnitude map, reducing isolated pixel noise and emphasizing coherent object movement. Second, we implement a

temporal band-pass filter across a sliding window of frames. This step is crucial as it suppresses low-frequency background drifts (*e.g.*, slow camera panning) and high-frequency erratic noise, thereby isolating the mid-to-high frequency motion characteristic of intentional actions. The motion saliency map $\mathbf{s}_{mot}$ highlights spatiotemporal regions with significant dynamic content.

**Dynamic Fusion.** As illustrated in Fig. 4 (c), the final spatiotemporal enhancement is modulated by a learnable gater. Since $\mathbf{s}_{mot}$ is derived at a pixel-level resolution while $\mathbf{s}_{spa}$ is extracted at the patch-level (DINOv3), these saliency maps exhibit heterogeneous spatial dimensions. To bridge this gap, we first perform individual normalization to ensure both signals are mapped into a unified $[0, 1]$ range. Subsequently, both $\mathbf{s}_{mot}$ and $\mathbf{s}_{spa}$ are aligned to the target visual patch size of the VLLM backbone (*e.g.*, $16 \times 16$ for Qwen3-VL (Bai et al., 2025a)) through bilinear interpolation. Conditioned on the video-query hidden states $\mathbf{H}$, the gating network employs a multi-layer perceptron (MLP) to learn a dynamic fusion weight $\beta \in [0, 1]$. The unified spatiotemporal weight is $\mathbf{w}_{sal} = \beta \cdot \mathbf{s}_{spa} + (1 - \beta) \cdot \mathbf{s}_{mot}$. The positive vision tokens $\mathbf{X}'_v$ are then enhanced via element-wise reweighting: $\mathbf{X}'_v = \mathbf{w}_{sal} \odot \mathbf{X}_v$, where $\odot$ denotes the Hadamard product. The enhanced vision tokens $\mathbf{X}'_v$ are fed back into the LLM decoder along with the original text tokens $\mathbf{X}_t$. The saliency-guided positive logit $q_t^p$ is obtained:

$$q_t^p = \text{logit}_\theta(y_t \mid \mathbf{X}'_v, \mathbf{X}_t, y_{<t}). \tag{9}$$

This mechanism ensures that the positive pass is anchored to visual evidence that is both spatially salient and temporally active, effectively establishing a robust "upper bound" for grounded reasoning. Alg. 2 defines the execution logic of the Saliency-Guided Enhancement. To enhance the precision of the generated saliency maps, several specialized techniques are employed within the algorithm:

- Adaptive Temporal Band-pass Filtering: To isolate meaningful motion from the raw magnitude $\mathbf{M}$, we implement a statistical filtering mechanism based on a sliding temporal window $W$.

  - Low-frequency Suppression ($\lambda_{low}$): By subtracting ($\mu + \lambda_{low}\sigma$), we eliminate low-magnitude motion components typically associated with background drift or slow camera panning.
  - High-frequency Removal ($\lambda_{high}$): An indicator function $\mathbb{I}(\mathbf{M}_i \leq \mu + \lambda_{high}\sigma)$ is used to suppress erratic, high-frequency noise caused by video artifacts or sudden camera shakes.

---

**Algorithm 2** Algorithm workflow of SGE

---

**Require:** Video Frames $\{V_i\}_{i=1}^K$; Query Tokens $\mathbf{X}_t$; Hidden States $\mathbf{H}$; Patch Size $P$.
**Ensure:** Enhanced Vision Tokens $\mathbf{X}'_v$; Positive Logit $q_p^t$.
1: $\mathbf{A} \leftarrow \text{Extract-Attn}(\text{DINOv3}(V))$    // Get CLS-to-patch attention
2: $\mathbf{s}_{spa} \leftarrow \text{Head-Agg}(\mathbf{A}_{[:,0,5:N+5]})$    // Filter out register tokens
3: **for** each consecutive pair $(V_{i-1}, V_i)$ **do**
4:    $\mathbf{F} \leftarrow \text{Farneback}(V_{i-1}^{gray}, V_i^{gray})$    // Estimate optical flow field
5:    $\mathbf{M}_i \leftarrow \sqrt{F_x^2 + F_y^2}$    // Compute raw motion magnitude
6:    $\mathbf{M}_i \leftarrow \text{Gaussian-Blur}(\mathbf{M}_i, \pi)$    // Spatial smoothing
7: **end for**
8: **for** each frame $i$ in window $W$ **do**
9:    $\mu, \sigma \leftarrow \text{Mean}(\mathbf{M}_W), \text{Std}(\mathbf{M}_W)$    // Compute temporal statistics
10:    $\mathbf{s}_{mot}^i \leftarrow \max(\mathbf{M}_i - (\mu + \lambda_{low}\sigma), 0)$    // Suppress background drift
11:    $\mathbf{s}_{mot}^i \leftarrow \mathbf{s}_{mot}^i \cdot \mathbb{I}(\mathbf{M}_i \leq \mu + \lambda_{high}\sigma)$    // Suppress high-freq noise
12: **end for**
13: $\bar{\mathbf{s}}_{spa}, \bar{\mathbf{s}}_{mot} \leftarrow \text{Normalize}(\mathbf{s}_{spa}, \mathbf{s}_{mot}) \in [0, 1]$
14: $\hat{\mathbf{s}}_{spa}, \hat{\mathbf{s}}_{mot} \leftarrow \text{Interpolate}(\bar{\mathbf{s}}_{spa}, \bar{\mathbf{s}}_{mot}, P \times P)$
15: $\beta \leftarrow \text{Sigmoid}(\text{MLP}(\text{Mean}(\mathbf{H})))$    // Dynamic gating
16: $\mathbf{w}_{sal} \leftarrow \beta \cdot \hat{\mathbf{s}}_{spa} + (1 - \beta) \cdot \hat{\mathbf{s}}_{mot}$    // Weighted fusion
17: $\mathbf{X}'_v \leftarrow \mathbf{w}_{sal} \odot \mathbf{X}_v$    // Element-wise token reweighting
18: $q_t^p \leftarrow \text{logit}_\theta(y_t \mid \mathbf{X}'_v, \mathbf{X}_t, y_{<t})$    // Generate positive logit

---

## C.5. Details of Optimization

We formulate the combined coordination of perturbation selection and saliency fusion as a Reinforcement Learning (RL) (Sutton et al., 1998; Li, 2017) problem. The primary motivation for adopting an RL framework is that the APC tool selection

is a non-differentiable operation. The discrete choice of specific perturbation tools from a dictionary $\mathcal{P}$ prevents the use of standard backpropagation (Rumelhart et al., 1986). By interacting with the frozen VLLM, both the APC and SGE learn to maximize suppressing hallucination pathways.

**APC Selection Policy (Discrete).** To accommodate the non-differentiable tool selection, we model the APC as a set of independent Bernoulli distributions (Casella & Berger, 2024). For each of the eight tools in $\mathcal{P}$, the router outputs a logit that is mapped via a sigmoid function to a selection probability. This Bernoulli formulation allows the model to sample multiple tools simultaneously, enabling the creation of complex negative samples that better capture compositional hallucinations.

**SGE Fusion Policy (Continuous).** The fusion weight $\beta$ for spatiotemporal enhancement is modeled as a continuous action. During training, we sample $\beta$ from a Normal distribution $\mathcal{N}(\mu, \sigma^2)$, where the mean $\mu$ is provided by the Gater and $\sigma$ is a fixed standard deviation to facilitate exploration. During inference, the mean value $\mu$ is used deterministically.

**Training Stability.** Given the high cost of VLLM inference, we implement gradient accumulation across $b$ steps to stabilize the policy updates. This effectively simulates a larger batch size for the RL agent, ensuring that the learned policies are robust across diverse video contexts. Additionally, we apply gradient clipping to prevent policy collapse during the early stages of spatiotemporal exploration. The complete training procedure is detailed in Alg. 3.

---

**Algorithm 3** OmniVCHall Training workflow

---

**Require:** Training set $\mathcal{D}$; Frozen VLLM $\theta$; Initialized Policy $\phi$; Baseline $B$; Accumulation Steps $b$; Prediction $y_{pred}$; Ground Truth $y_{gt}$; Decay Factor $\varphi$; Learning Rate $l$; Gradient Clip Threshold $\eta$.
**Ensure:** Optimized Parameters $\phi$.

1: $B \leftarrow 0, \quad step \leftarrow 0, \quad \nabla\mathcal{L} \leftarrow 0$
2: **for** each batch $(y_{pred}, y_{gt}) \in \mathcal{D}$ **do**
3:     $R \leftarrow 1.0$ **if** $y_{pred} = y_{gt}$ **else** $-1.0$
4:     $B \leftarrow \varphi \cdot B + (1 - \varphi) \cdot R$
5:     $Adv \leftarrow R - B$    // Calculate advantage signal
6:     $\mathcal{L} \leftarrow -(\sum \log P(\mathbf{a}_{apc}) + \log P(\mathbf{a}_{sge})) \cdot Adv$
7:     $\nabla\mathcal{L} \leftarrow \nabla\mathcal{L} + \text{Backprop}(\mathcal{L})/M$    // Accumulate
8:     $step \leftarrow step + 1$
9:     **if** $step \pmod M = 0$ **then**
10:       $\phi \leftarrow \text{OptimizerStep}(\phi, \nabla\mathcal{L}, l, \text{clip} = \eta)$    // Update
11:       $\nabla\mathcal{L} \leftarrow 0$
12:     **end if**
13: **end for**

---

# D. Experimental Details

## D.1. Model Details

We evaluated 39 VLLMs from 13 different model families, including ten open-source model families: Qwen3-VL (Bai et al., 2025a), Qwen2.5-VL (Bai et al., 2025b), LLaVA-NeXT-Video (Zhang et al., 2024b), VideoLLaMA3 (Zhang et al., 2025), GLM series (Hong et al., 2025), Molmo2 (Clark et al., 2026), Kimi-VL (Team et al., 2025), InternVL3.5 (Wang et al., 2025), VideoChat-Flash (Li et al., 2024c), MiniCPM-V series (Yu et al., 2025; Yao et al., 2025), and three close-source model families: GPT series (OpenAI, a;b;c), Gemini series (DeepMind, a;b;c), Doubao series (Volcengine, a;b). These models represent a wide variety of architectural designs and training paradigms. Additionally, we included a human baseline. All models and their checkpoints are listed in Tab. 13. The open-source models are available via HuggingFace (https://huggingface.co/), while the proprietary models are accessible through their respective providers' APIs. All evaluations for the proprietary models were conducted in January 2026.

*Table 13.* Details on model names and corresponding checkpoints.

| Model | Size | Checkpoint | API |
|---|---|---|---|
| *Open-source Models* | | | |
| Qwen3-VL-Instruct | 235B | https://huggingface.co/Qwen/Qwen3-VL-235B-A22B-Instruct | |
| | 32B | https://huggingface.co/Qwen/Qwen3-VL-32B-Instruct | |
| | 8B | https://huggingface.co/Qwen/Qwen3-VL-8B-Instruct | |
| | 4B | https://huggingface.co/Qwen/Qwen3-VL-4B-Instruct | |
| | 2B | https://huggingface.co/Qwen/Qwen3-VL-2B-Instruct | |
| Qwen3-VL-Thinking | 235B | https://huggingface.co/Qwen/Qwen3-VL-235B-A22B-Thinking | https://www.aliyun.com |
| | 32B | https://huggingface.co/Qwen/Qwen3-VL-32B-Thinking | |
| | 8B | https://huggingface.co/Qwen/Qwen3-VL-8B-Thinking | |
| | 4B | https://huggingface.co/Qwen/Qwen3-VL-4B-Thinking | |
| | 2B | https://huggingface.co/Qwen/Qwen3-VL-2B-Thinking | |
| Qwen2.5-VL-Instruct | 32B | https://huggingface.co/Qwen/Qwen2.5-VL-32B-Instruct | |
| | 7B | https://huggingface.co/Qwen/Qwen2.5-VL-7B-Instruct | |
| | 3B | https://huggingface.co/Qwen/Qwen2.5-VL-3B-Instruct | |
| LLaVA-NeXT-Video | 34B | https://huggingface.co/llava-hf/LLaVA-NeXT-Video-34B-hf | - |
| | 7B | https://huggingface.co/llava-hf/LLaVA-NeXT-Video-7B-hf | |
| VideoLLaMA3 | 7B | https://huggingface.co/DAMO-NLP-SG/VideoLLaMA3-7B | - |
| | 2B | https://huggingface.co/DAMO-NLP-SG/VideoLLaMA3-2B | |
| GLM-4.6v | 108B | https://huggingface.co/zai-org/GLM-4.6V | |
| GLM-4.6v-flash | 108B | https://huggingface.co/zai-org/GLM-4.6V-Flash | https://bigmodel.cn |
| GLM-4.5v | 108B | https://huggingface.co/zai-org/GLM-4.5V | |
| Molmo2 | 8B | https://huggingface.co/allenai/Molmo2-8B | - |
| | 4B | https://huggingface.co/allenai/Molmo2-4B | |
| Kimi-VL-Thinking | 16B | https://huggingface.co/moonshotai/Kimi-VL-A3B-Thinking-2506 | https://platform.moonshot.ai |
| Kimi-VL-Instruct | 16B | https://huggingface.co/moonshotai/Kimi-VL-A3B-Instruct | |
| InternVL3.5 | 30B | https://huggingface.co/OpenGVLab/InternVL3_5-30B-A3B | |
| | 8B | https://huggingface.co/OpenGVLab/InternVL3_5-8B | - |
| | 4B | https://huggingface.co/OpenGVLab/InternVL3_5-4B | |
| VideoChat-Flash | 7B | https://huggingface.co/OpenGVLab/VideoChat-Flash-Qwen2-7B_res448 | - |
| | 2B | https://huggingface.co/OpenGVLab/VideoChat-Flash-Qwen2_5-2B_res448 | |
| MiniCPM-V-4.5 | 8B | https://huggingface.co/openbmb/MiniCPM-V-4_5 | - |
| MiniCPM-V-4 | 4B | https://huggingface.co/openbmb/MiniCPM-V-4 | |
| *Proprietary Models* | | | |
| GPT-5.2 | - | - | |
| GPT-5 | - | - | https://openai.com |
| GPT-4o | - | - | |
| Gemini-3-pro | - | - | |
| Gemini-2.5-pro | - | - | https://ai.google.dev |
| Gemini-2.5-flash | - | - | |
| Doubao-Seed-1.8 | - | - | https://console.volcengine.com |
| Doubao-Seed-1.6 | - | - | |

## D.2. Metrics Details

We adopt Accuracy as the primary evaluation metric for both YNQA and MCQA tasks. It is formulated as:

$$Acc = \frac{1}{n} \sum_{i=1}^{n} \mathbb{I}(y_{pred}^i = y_{gt}^i) \tag{10}$$

where $n$ denotes the total number of test queries. $y_{pred}^i$ and $y_{gt}^i$ represent the predicted and ground-truth labels respectively. $\mathbb{I}(\cdot)$ is the indicator function. This unified metric allows for a consistent and direct comparison across different testing types. We also provide the following granular metrics in OmniVCHall.

**YNQA Subset Accuracy.** To identify potential linguistic biases, such as a "Yes-bias", we calculate accuracy independently for affirmative and negative queries: $YesAcc = \frac{1}{n_{yes}} \sum_{i \in \mathcal{I}_{yes}} \mathbb{I}(y_{pred}^i = y_{gt}^i)$, and $NoAcc = \frac{1}{n_{no}} \sum_{i \in \mathcal{I}_{no}} \mathbb{I}(y_{pred}^i = y_{gt}^i)$, where $n_{yes}$ and $n_{no}$ represent the number of queries where the ground truth is "Yes" and "No", respectively.

**MCQA Discriminative Metrics.** For multiple-choice tasks, we employ Macro-averaged Precision, Recall, and F1-score to assess the model's discriminative power across all choice categories. By treating each option (*e.g.*, A, B, and C) as an independent class, these metrics prevent performance from being masked by simple option biases:

- **Macro-Precision** ($P_{macro} = \frac{1}{K} \sum_{i=1}^{K} \frac{TP_i}{TP_i + FP_i}$): Measures the average exactness of the model in identifying the correct spatiotemporal description across all $K$ candidate categories.

- **Macro-Recall** ($R_{macro} = \frac{1}{K} \sum_{i=1}^{K} \frac{TP_i}{TP_i + FN_i}$): Evaluates the model's average ability to correctly retrieve the grounded answer from each specific option pool, ensuring robust detection of adversarial distractors.

- **Macro-F1-Score** ($F1_{macro} = \frac{1}{K} \sum_{i=1}^{K} \frac{2 \cdot P_i \cdot R_i}{P_i + R_i}$): Provides a balanced harmonic mean of precision and recall across all classes, ensuring the evaluation is not skewed by the frequency of adversarial injections.

---

### System Prompt
You are a professional AI Video Assistant specialized in complex spatiotemporal reasoning and hallucination detection. Your objective is to analyze the provided video content and respond to the associated question while strictly adhering to the following logic and constraints:

### Task Instructions
1. **Core Task**:
   - If the task is **Yes/No Question-Answering (YNQA)**: Your answer must be strictly limited to "Yes" or "No".
   - If the task is **Multi-Choice Question-Answering (MCQA)**: Your answer must be a single identifier selected from the provided options (e.g., "A", "B", or "C").
2. **Reasoning Requirement**: You must provide a comprehensive reasoning process that justifies your answer. This rationale should be grounded in specific spatiotemporal evidence from the video, such as object movements, event sequences, or attribute changes.
3. **Constraint**: Do not include any conversational filler, introductory remarks, or additional text outside of the specified format.

### Input Data
- **Question**: *{Question}*
- **Options (for MCQA only)**: *{Options}*

### Output Format
You must strictly return the results in the following JSON schema:
{
    "answer": "[Selected Option or Yes/No]",
    "reasoning": "[Detailed spatiotemporal reasoning process]"
}

*Figure 17.* Prompt for testing models. *Terms* (*e.g.*, {*Question*} and {*Options*}) represent the dynamic input.

*Table 14.* Hyperparameter settings.

| Hyperparameter | Value |
|---|---|
| Cumulative probability threshold ($\gamma$) | 0.4 |
| Logit weight ($\alpha_1$) | 0.8 |
| Logit weight ($\alpha_2$) | 0.4 |
| Patch size ($P$) | 16/14 |
| Standard deviation of spatial Gaussian filter ($\pi$) | 1.0 |
| Temporal window size ($W$) | 5 |
| Low-frequency threshold ($\lambda_{low}$) | 0.1 |
| High-frequency threshold ($\lambda_{high}$) | 0.9 |
| Standard deviation of distribution in SGE training ($\sigma$) | 0.1 |
| Decay factor ($\varphi$) | 0.95 |
| Gradient clip threshold ($\eta$) | 1.0 |
| Learning rate ($l$) | $1 \times 10^{-4}$ |
| Batch size ($b$) | 32 |
| Random seed | 2025 |

### D.3. Implementation Details

All experiments were conducted using 8 NVIDIA RTX A6000 48GB GPUs. We adhered to the inference and model hyperparameters outlined in the respective original models, and employed greedy decoding during generation for a fair comparison. We utilized a unified prompt template throughout the experimental phase, as shown in Fig. 17. To evaluate the efficacy of our proposed TriCD, we repurpose the OmniVCHall benchmark into a supervised training and evaluation suite. Specifically, we partition the 823 videos and their corresponding 9,027 QA pairs into a 7:2:1 split for training, validation, and testing, respectively. This partition ensures that the performance gains are evaluated on unseen spatiotemporal scenarios. We implement TriCD on two representative backbones: VideoLLaMA3-7B (Zhang et al., 2025) and Qwen3-VL-Instruct-8B

([Bai et al., 2025a](#)), to demonstrate its scalability across different model capacities. We use a fixed predefined randomization seed across experiments. During the optimization phase of TriCD, the VLLM backbone is kept strictly frozen, focusing solely on the lightweight APC and SGE modules. These are optimized using the REINFORCE algorithm with a learning rate of $l = 1 \times 10^{-4}$. To stabilize the reinforcement learning process, we utilize a gradient accumulation of $b = 32$ steps and a moving baseline with a decay factor of $\varphi = 0.95$. For the APC module, the cumulative probability threshold for dynamic tool selection is set to $\gamma = 0.4$. Spatial saliency in the SGE module is extracted using DINOv3 (dinov3-vitl16-pretrain-lvd1689m), while temporal motion features are computed via the Farneback algorithm with a standard window size $W$ of 5. The hyperparameter settings are shown in Tab. 14.

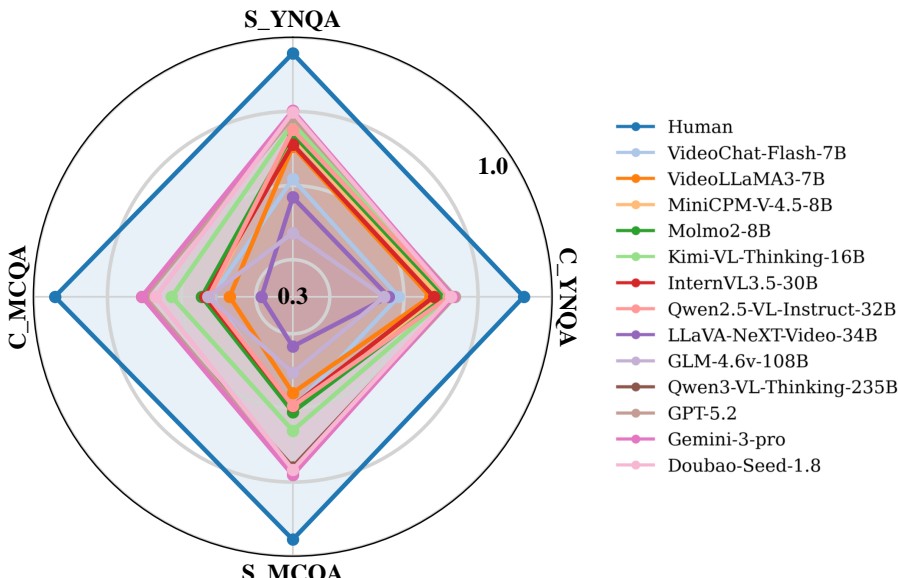

*Figure 18.* Comparative accuracy radar chart of representative VLLMs across the four sub-tasks. The outer boundary represents human performance, highlighting the gap in spatiotemporal grounding for leading models from each architectural family.

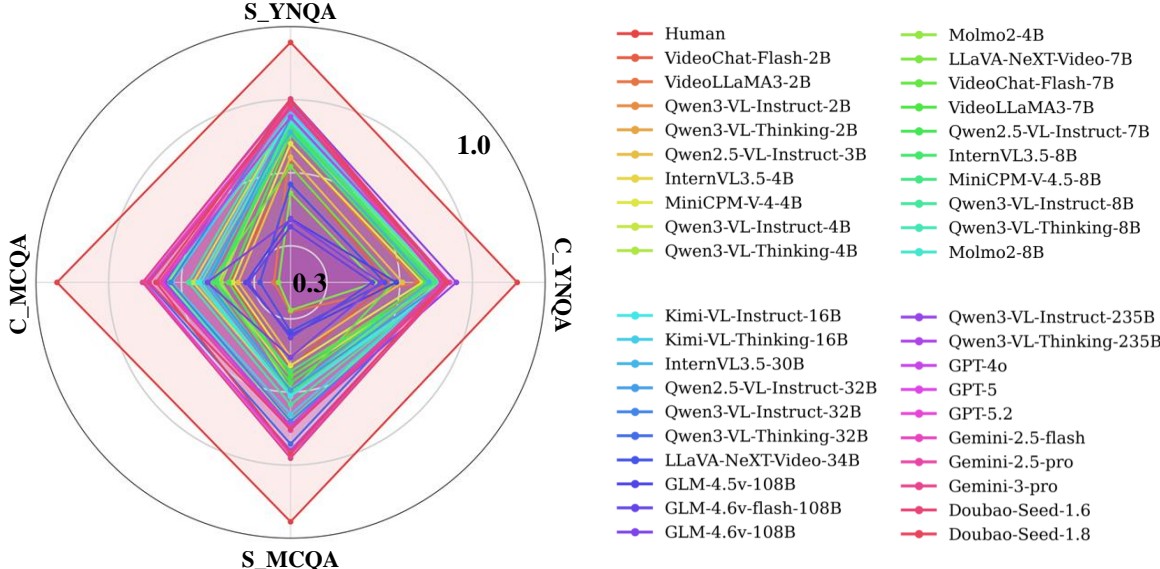

*Figure 19.* Holistic accuracy distribution of all 39 evaluated models on OmniVCHall. The dense overlapping patterns reveal collective model vulnerabilities across simple and complex reasoning tasks relative to the near-perfect human baseline.

## D.4. Benchmark Results

Tab. 17 provides the exhaustive accuracy for all evaluated VLLMs across the four sub-tasks in both real-world and AI-generated domains. To facilitate a deeper diagnostic of model behavior, Tab. 18 and Tab. 19 report supplementary performance statistics, including class-specific accuracy for YNQA and Macro Precision, Recall, and F1-score for MCQA, under real-world and generated scenarios, respectively. These comprehensive tables offer a granular view of architectural robustness and potential linguistic biases across the heterogeneous video sources in OmniVCHall. Furthermore, Fig. 18 and Fig. 19 visualize the accuracy distribution across the four task dimensions, comparing representative model families and the full model suite, respectively, against the stable human performance ceiling.

## D.5. Baseline Methods

In this section, we provide a detailed description of baseline methods:

- **MotionCD** (Kong et al., 2025): This method utilizes motion-aware contrastive decoding to penalize hallucinations by corrupting temporal dynamics in the video. We follow the official implementation available at `https://github.com/Stevetich/EventHallusion`.

- **TCD** (Zhang et al., 2024a): This method focuses on mitigating event-level hallucinations by establishing a temporal contrastive baseline through frame shuffling. We follow the official implementation available at `https://github.com/xzhouzeng/MHBench`.

- **DINO-HEAL** (Li et al., 2025a): This method employs spatial saliency extracted from DINOv2 features to enhance visual grounding and reduce hallucinations. We follow the official implementation available at `https://github.com/CyL97/VidHalluc`.

## D.6. Optimization and Efficiency

To facilitate high-efficiency training, we pre-compute and locally store the tool embeddings and saliency maps, significantly reducing the redundant computational overhead during the RL phase. In the SGE module, saliency computation is strictly restricted to the frames selected by the VLLM's processor to minimize unnecessary visual processing. Furthermore, to maintain high throughput, TriCD utilizes a parallel branching strategy where the SGE and APC modules independently process shared hidden states from the frozen backbone to generate positive and negative logit adjustments. These concurrent outputs are then integrated into a single contrastive decoding step, effectively mitigating hallucinations without introducing a sequential bottleneck or significantly increasing inference latency. We employ a dual-stage spatiotemporal perturbation pipeline: global temporal tools (*e.g.*, temporal sampling) are first applied to the entire video sequence to disrupt long-range dynamics, followed by local spatial tools (*e.g.*, grayscale transformation) that are restricted to the specific frames indexed by the VLLM's processor. This hierarchical approach ensures that negative samples incorporate both coarse-grained temporal inconsistencies and fine-grained spatial distortions while maintaining computational efficiency by avoiding redundant processing on unsampled frames.

As detailed in Tab. 15, we evaluate the computational efficiency of TriCD compared to existing baselines. Despite performing a dual-boundary calibration (APC and SGE), TriCD maintains a manageable inference time. While sequential methods like MotionCD (8.05s) and TCD (7.95s) nearly double the backbone's latency (3.89s) due to independent dual-pass executions, our parallel branching strategy ensures that the positive and negative adjustments are computed concurrently from shared hidden states. This architectural choice effectively prevents a sequential bottleneck.

*Table 15.* Inference efficiency on a single NVIDIA RTX A6000 (48GB) GPU. We report the average time per query for Qwen3-VL-Instruct-8B using different methods.

| Model | Average Time | Average Accuracy |
|---|---|---|
| Qwen3-VL-Instruct-8B | 3.89s | 0.67 |
| +MotionCD | 8.05s | 0.68 |
| +TCD | 7.95s | 0.69 |
| +DINO-HEAL | 4.65s | 0.68 |
| TriCD | 9.12s | 0.73 |

## D.7. Hyperparameter Analysis

This additive refinement structure provides a robust theoretical foundation for multi-source logit calibration by decoupling the dual forces of evidence enhancement and bias suppression. Specifically, the positive grounding residual $(q_t^p - q_t^o)$ isolates the semantic gain achieved by focusing the visual encoder on spatiotemporal regions identified as salient. It effectively rewards tokens supported by verified visual evidence to correct for oversight errors in the original pass. Simultaneously, the negative suppression residual $(q_t^o - q_t^n)$ captures the pure hallucination bias that is the disparity between the model's standard reasoning and its over-reliance on linguistic priors. By pivoting these residuals around the baseline $q_t^o$, the formulation enables the independent calibration of grounding intensity $(\alpha_1)$ and suppression strength $(\alpha_2)$. This flexibility ensures that TriCD can be fine-tuned to the specific inductive biases of different VLLM backbones, maintaining a stable decision boundary even in complex video scenarios.

As shown in Fig. 8 and Tab. 16 confirms that the optimal balance is achieved at $\alpha_1 = 0.8$ and $\alpha_2 = 0.4$.

*Table 16.* Hyperparameter sensitivity analysis of $\alpha_1$ and $\alpha_2$.

| $\alpha_2 \setminus \alpha_1$ | 0.0 | 0.2 | 0.4 | 0.6 | 0.8 | 1.0 |
|---|---|---|---|---|---|---|
| 0.0 | 0.67 | 0.67 | 0.68 | 0.68 | 0.69 | 0.68 |
| 0.2 | 0.67 | 0.68 | 0.69 | 0.71 | 0.72 | 0.69 |
| 0.4 | 0.68 | 0.68 | 0.71 | 0.73 | **0.73** | 0.72 |
| 0.6 | 0.69 | 0.69 | 0.70 | 0.72 | 0.72 | 0.71 |
| 0.8 | 0.69 | 0.70 | 0.70 | 0.71 | 0.72 | 0.70 |
| 1.0 | 0.71 | 0.70 | 0.69 | 0.70 | 0.70 | 0.69 |

## D.8. Case Study

To qualitatively evaluate the effectiveness of TriCD, we present two representative cases.

In the YNQA task (Fig. 20), the base model fails to perceive the global camera motion, providing an incorrect "No" response. By contrast, TriCD correctly identifies the panning direction. Similarly, in the MCQA task (Fig. 21), the backbone incorrectly hallucinates a "Tray" in the boy's hand. TriCD successfully avoids this distractor and selects the adversarial option "C. All others are wrong". This demonstrates that the synergy between our adaptive perturbations and dual-saliency grounding, incorporating both DINOv3 spatial features and motion cues, enables the model to maintain strict visual grounding even in complex, cluttered real-world environments.

Other cases are shown in Figs. 22 - 26.

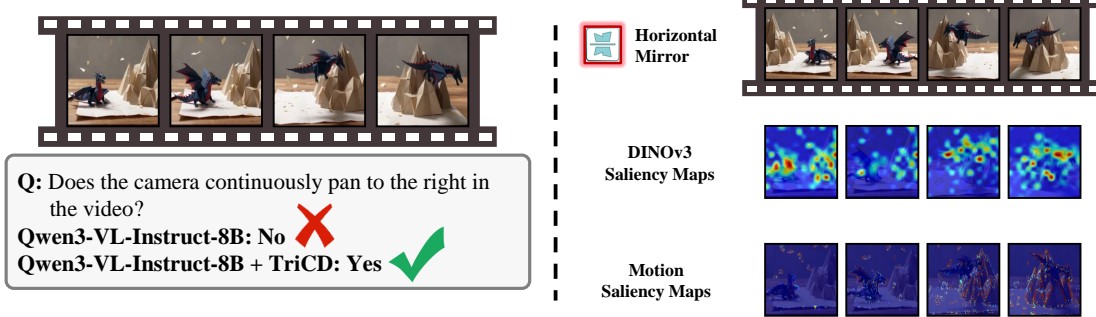

*Figure 20.* Visualization of TriCD's robustness in the YNQA task.

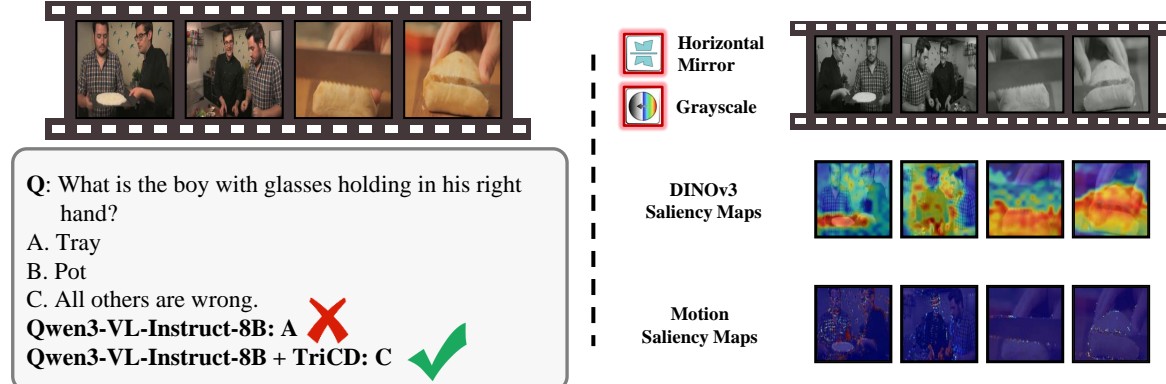

*Figure 21.* Visualization of TriCD's robustness in the MCQA task.

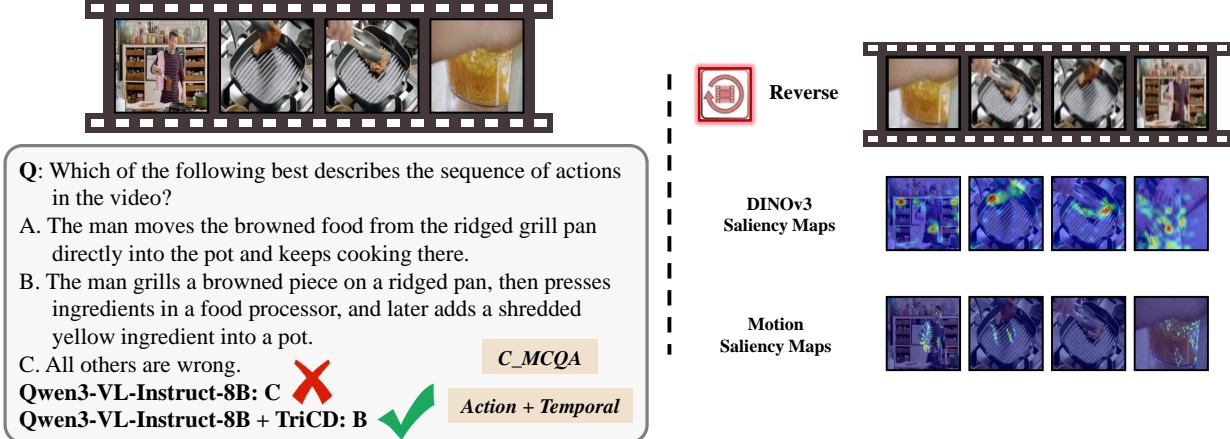

*Figure 22.* Case study of a compositional Action + Temporal question.

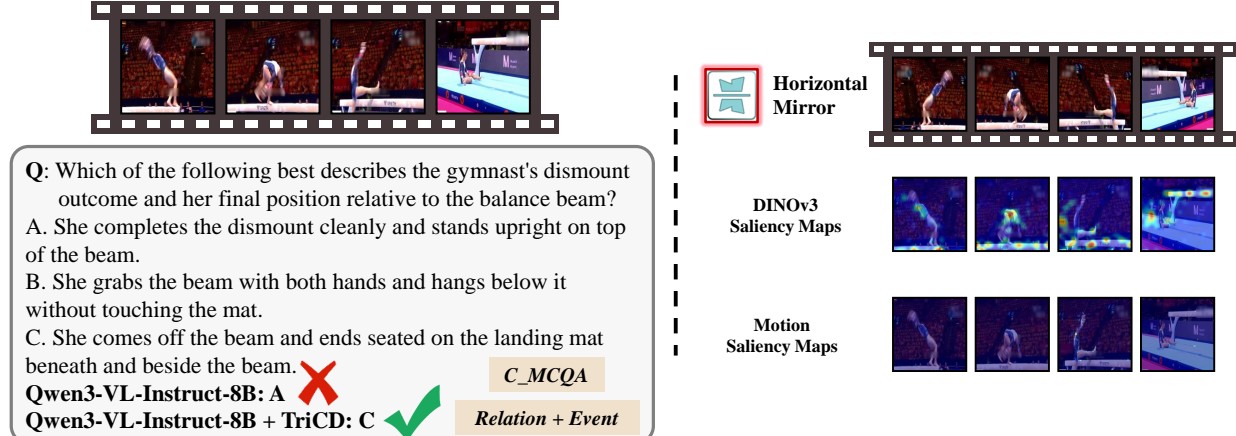

*Figure 23.* Case study of a compositional Relation + Event question.

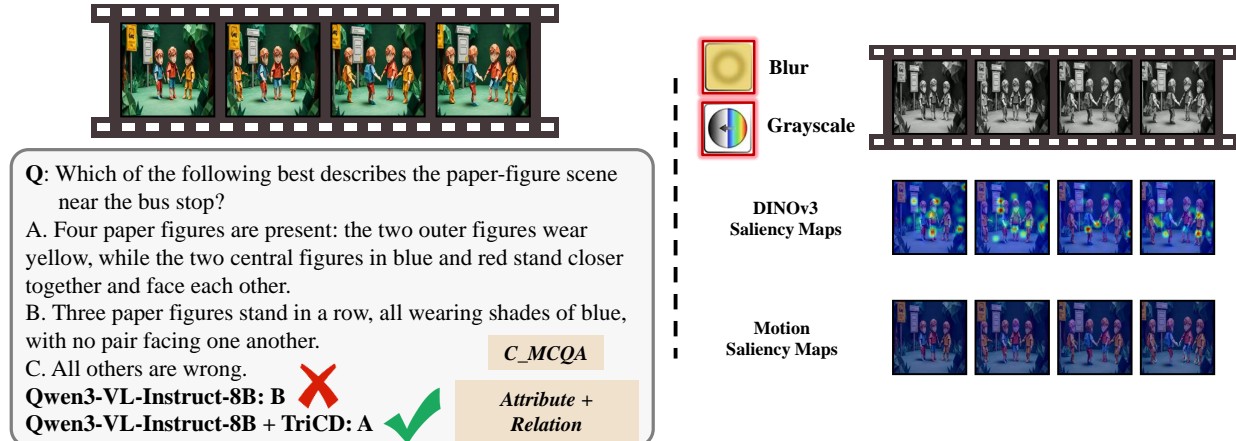

Figure 24. Case study of a compositional Attribute + Relation question.

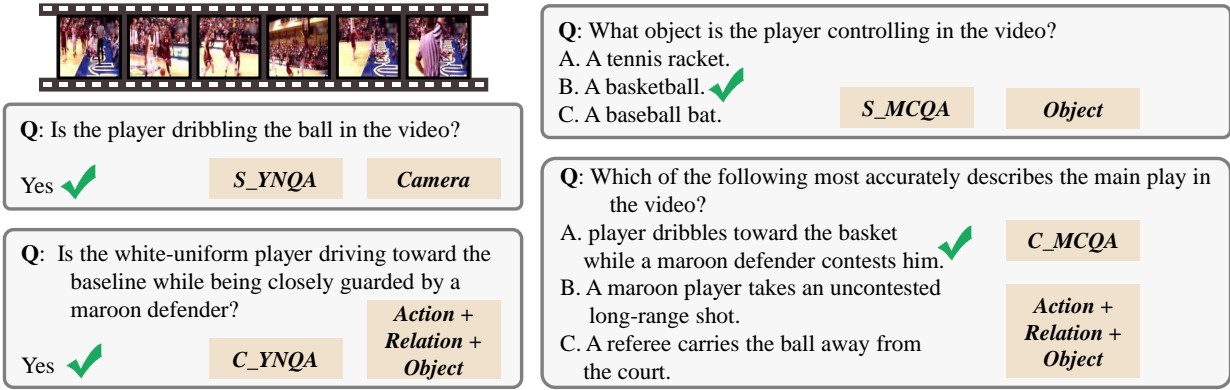

Figure 25. Case study on a sports video. The single questions rely on one dominant evidence type, such as Camera or Object, whereas the compositional questions require the joint support of multiple types, e.g., Action + Relation + Object. This example illustrates that compositionality is determined by jointly necessary evidence, not by question length or surface complexity alone.

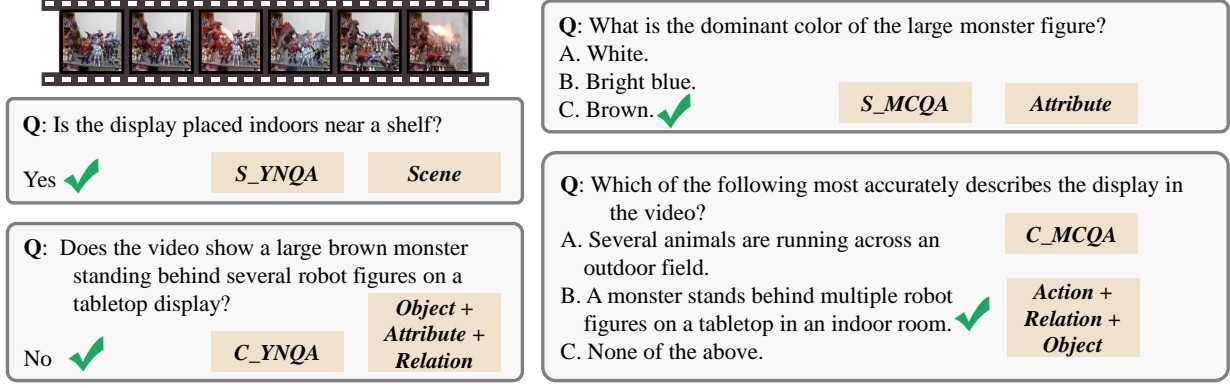

Figure 26. Case study on an indoor display video. The single questions can be resolved from a single evidence type, such as Scene or Attribute, while the compositional questions require multiple jointly necessary types, such as Object + Attribute + Relation or Action + Relation + Object. The example highlights how the compositional label captures cross-type dependence rather than merely more detailed wording.

*Table 17.* Main benchmark results on accuracy. The gray shaded cells highlight the best and second-best results in the open-source and commercial models, respectively. The ***best model result*** among all models is highlighted. Avg means average.

| Model | Size | Real-world | | | | | AI-generated | | | | | Avg |
|---|---|---|---|---|---|---|---|---|---|---|---|---|
| | | S_YNQA | C_YNQA | S_MCQA | C_MCQA | Avg | S_YNQA | C_YNQA | S_MCQA | C_MCQA | Avg | |
| Human | - | **0.96** | **0.93** | **0.95** | **0.95** | **0.95** | **0.96** | **0.91** | **0.96** | **0.93** | **0.94** | **0.95** |
| *Open-source Models* | | | | | | | | | | | | |
| VideoChat-Flash (Li et al., 2024c) | 2B | 0.61 | 0.56 | 0.53 | 0.45 | 0.55 | 0.64 | 0.57 | 0.54 | 0.48 | 0.57 | 0.56 |
| VideoLLaMA3 (Zhang et al., 2025) | 2B | 0.62 | 0.63 | 0.36 | 0.32 | 0.51 | 0.73 | 0.67 | 0.40 | 0.37 | 0.58 | 0.54 |
| Qwen3-VL-Instruct (Bai et al., 2025a) | 2B | 0.68 | 0.66 | 0.50 | 0.46 | 0.59 | 0.75 | 0.71 | 0.59 | 0.54 | 0.67 | 0.63 |
| Qwen3-VL-Thinking (Bai et al., 2025a) | 2B | 0.66 | 0.64 | 0.59 | 0.55 | 0.62 | 0.75 | 0.72 | 0.64 | 0.56 | 0.69 | 0.65 |
| Qwen2.5-VL-Instruct (Bai et al., 2025b) | 3B | 0.59 | 0.60 | 0.45 | 0.40 | 0.52 | 0.68 | 0.62 | 0.56 | 0.49 | 0.61 | 0.57 |
| InternVL3.5 (Wang et al., 2025) | 4B | 0.67 | 0.67 | 0.55 | 0.46 | 0.61 | 0.74 | 0.72 | 0.56 | 0.53 | 0.66 | 0.64 |
| MiniCPM-V-4 (Yu et al., 2025) | 4B | 0.64 | 0.64 | 0.52 | 0.44 | 0.57 | 0.71 | 0.68 | 0.54 | 0.48 | 0.63 | 0.60 |
| Qwen3-VL-Instruct (Bai et al., 2025a) | 4B | 0.71 | 0.66 | 0.59 | 0.55 | 0.64 | 0.81 | 0.77 | 0.64 | 0.58 | 0.72 | 0.68 |
| Qwen3-VL-Thinking (Bai et al., 2025a) | 4B | 0.69 | 0.67 | 0.67 | 0.63 | 0.67 | 0.79 | 0.74 | 0.69 | 0.65 | 0.73 | 0.70 |
| Molmo2 (Clark et al., 2026) | 4B | 0.70 | 0.68 | 0.59 | 0.47 | 0.63 | 0.78 | 0.72 | 0.59 | 0.51 | 0.68 | 0.66 |
| LLaVA-NeXT-Video (Zhang et al., 2024b) | 7B | 0.52 | 0.55 | 0.33 | 0.29 | 0.44 | 0.57 | 0.52 | 0.42 | 0.38 | 0.49 | 0.47 |
| VideoChat-Flash (Li et al., 2024c) | 7B | 0.61 | 0.59 | 0.58 | 0.49 | 0.58 | 0.62 | 0.58 | 0.62 | 0.55 | 0.60 | 0.59 |
| VideoLLaMA3 (Zhang et al., 2025) | 7B | 0.65 | 0.64 | 0.55 | 0.46 | 0.59 | 0.75 | 0.70 | 0.57 | 0.48 | 0.65 | 0.62 |
| Qwen2.5-VL-Instruct (Bai et al., 2025b) | 7B | 0.67 | 0.62 | 0.56 | 0.49 | 0.60 | 0.77 | 0.75 | 0.59 | 0.52 | 0.69 | 0.64 |
| InternVL3.5 (Wang et al., 2025) | 8B | 0.67 | 0.68 | 0.52 | 0.48 | 0.60 | 0.75 | 0.73 | 0.56 | 0.52 | 0.67 | 0.64 |
| MiniCPM-V-4.5 (Yu et al., 2025) | 8B | 0.69 | 0.69 | 0.59 | 0.51 | 0.64 | 0.76 | 0.70 | 0.62 | 0.54 | 0.68 | 0.66 |
| Qwen3-VL-Instruct (Bai et al., 2025a) | 8B | 0.72 | 0.67 | 0.60 | 0.55 | 0.65 | 0.82 | 0.77 | 0.66 | 0.61 | 0.74 | 0.70 |
| Qwen3-VL-Thinking (Bai et al., 2025a) | 8B | 0.68 | 0.68 | 0.65 | 0.64 | 0.67 | 0.79 | 0.76 | 0.71 | 0.65 | 0.74 | 0.71 |
| Molmo2 (Clark et al., 2026) | 8B | 0.70 | 0.69 | 0.60 | 0.53 | 0.65 | 0.77 | 0.73 | 0.62 | 0.56 | 0.69 | 0.67 |
| Kimi-VL-Instruct (Team et al., 2025) | 16B | 0.70 | 0.68 | 0.60 | 0.54 | 0.64 | 0.81 | 0.78 | 0.63 | 0.57 | 0.72 | 0.68 |
| Kimi-VL-Thinking (Team et al., 2025) | 16B | 0.74 | 0.69 | 0.64 | 0.62 | 0.68 | 0.79 | 0.75 | 0.68 | 0.63 | 0.73 | 0.71 |
| InternVL3.5 (Wang et al., 2025) | 30B | 0.67 | 0.66 | 0.59 | 0.53 | 0.62 | 0.74 | 0.70 | 0.60 | 0.54 | 0.67 | 0.65 |
| Qwen2.5-VL-Instruct (Bai et al., 2025b) | 32B | 0.69 | 0.69 | 0.55 | 0.50 | 0.62 | 0.79 | 0.75 | 0.64 | 0.55 | 0.71 | 0.67 |
| Qwen3-VL-Instruct (Bai et al., 2025a) | 32B | 0.73 | 0.70 | 0.66 | 0.62 | 0.69 | 0.84 | 0.77 | 0.7 | 0.64 | 0.76 | 0.72 |
| Qwen3-VL-Thinking (Bai et al., 2025a) | 32B | 0.73 | 0.67 | 0.73 | 0.70 | 0.71 | 0.82 | 0.76 | 0.76 | 0.68 | 0.77 | 0.74 |
| LLaVA-NeXT-Video (Zhang et al., 2024b) | 34B | 0.54 | 0.58 | 0.38 | 0.33 | 0.47 | 0.59 | 0.54 | 0.49 | 0.44 | 0.53 | 0.50 |
| GLM-4.5v (Hong et al., 2025) | 108B | 0.46 | 0.59 | 0.43 | 0.39 | 0.47 | 0.48 | 0.59 | 0.46 | 0.45 | 0.50 | 0.49 |
| GLM-4.6v-flash (Hong et al., 2025) | 108B | 0.43 | 0.53 | 0.42 | 0.35 | 0.44 | 0.47 | 0.52 | 0.49 | 0.48 | 0.49 | 0.46 |
| GLM-4.6v (Hong et al., 2025) | 108B | 0.46 | 0.54 | 0.47 | 0.51 | 0.49 | 0.48 | 0.55 | 0.54 | 0.55 | 0.52 | 0.51 |
| Qwen3-VL-Instruct (Bai et al., 2025a) | 235B | 0.75 | *0.73* | 0.69 | 0.64 | 0.71 | *0.84* | *0.79* | 0.71 | 0.65 | 0.77 | 0.74 |
| Qwen3-VL-Thinking (Bai et al., 2025a) | 235B | 0.75 | 0.70 | 0.75 | 0.73 | 0.73 | 0.84 | 0.77 | 0.77 | *0.68* | *0.78* | 0.76 |
| *Proprietary Models* | | | | | | | | | | | | |
| GPT-4o (OpenAI, a) | - | 0.72 | 0.69 | 0.71 | 0.66 | 0.70 | 0.78 | 0.76 | 0.67 | 0.64 | 0.73 | 0.71 |
| GPT-5 (OpenAI, b) | - | 0.72 | 0.71 | 0.70 | 0.72 | 0.72 | 0.81 | 0.76 | 0.70 | 0.66 | 0.75 | 0.73 |
| GPT-5.2 (OpenAI, c) | - | 0.74 | 0.71 | 0.77 | 0.72 | 0.74 | 0.82 | 0.77 | 0.77 | 0.68 | 0.77 | 0.76 |
| Gemini-2.5-flash (DeepMind, a) | - | 0.74 | 0.68 | 0.70 | 0.69 | 0.71 | 0.82 | 0.76 | 0.70 | 0.64 | 0.75 | 0.73 |
| Gemini-2.5-pro (DeepMind, b) | - | 0.74 | 0.68 | 0.71 | 0.68 | 0.71 | 0.81 | 0.75 | 0.70 | 0.64 | 0.74 | 0.73 |
| Gemini-3-pro (DeepMind, c) | - | *0.78* | 0.71 | *0.79* | *0.74* | *0.76* | 0.82 | 0.76 | *0.77* | 0.67 | 0.77 | *0.77* |
| Doubao-Seed-1.6 (Volcengine, a) | - | 0.73 | 0.70 | 0.71 | 0.71 | 0.71 | 0.82 | 0.76 | 0.69 | 0.67 | 0.75 | 0.73 |
| Doubao-Seed-1.8 (Volcengine, b) | - | 0.76 | 0.70 | 0.76 | 0.67 | 0.73 | 0.82 | 0.76 | 0.77 | 0.67 | 0.77 | 0.75 |

*Table 18.* Main results on other metrics under the real-word scenario.

| Model | Size | S_YNQA | | C_YNQA | | S_MCQA | | | C_MCQA | | |
|---|---|---|---|---|---|---|---|---|---|---|---|
| | | YesAcc | NoAcc | YesAcc | NoAcc | Macro Precision | Macro Recall | Macro F1 | Macro Precision | Macro Recall | Macro F1 |
| *Open-source Models* | | | | | | | | | | | |
| VideoChat-Flash (Li et al., 2024c) | 2B | 0.89 | 0.42 | 0.89 | 0.32 | 0.54 | 0.53 | 0.53 | 0.45 | 0.45 | 0.45 |
| VideoLLaMA3 (Zhang et al., 2025) | 2B | 0.62 | 0.62 | 0.62 | 0.65 | 0.46 | 0.34 | 0.23 | 0.34 | 0.31 | 0.19 |
| Qwen3-VL-Instruct (Bai et al., 2025a) | 2B | 0.78 | 0.62 | 0.68 | 0.63 | 0.51 | 0.49 | 0.49 | 0.47 | 0.45 | 0.45 |
| Qwen3-VL-Thinking (Bai et al., 2025a) | 2B | 0.74 | 0.61 | 0.65 | 0.64 | 0.60 | 0.58 | 0.57 | 0.57 | 0.55 | 0.54 |
| Qwen2.5-VL-Instruct (Bai et al., 2025b) | 3B | 0.88 | 0.40 | 0.87 | 0.40 | 0.54 | 0.43 | 0.41 | 0.37 | 0.40 | 0.46 |
| InternVL3.5 (Wang et al., 2025) | 4B | 0.70 | 0.65 | 0.59 | 0.73 | 0.56 | 0.55 | 0.55 | 0.46 | 0.46 | 0.46 |
| MiniCPM-V-4 (Yu et al., 2025) | 4B | 0.88 | 0.48 | 0.83 | 0.51 | 0.53 | 0.52 | 0.52 | 0.46 | 0.44 | 0.44 |
| Qwen3-VL-Instruct (Bai et al., 2025a) | 4B | 0.63 | 0.77 | 0.48 | 0.73 | 0.59 | 0.58 | 0.58 | 0.56 | 0.55 | 0.55 |
| Qwen3-VL-Thinking (Bai et al., 2025a) | 4B | 0.66 | 0.71 | 0.50 | 0.78 | 0.68 | 0.67 | 0.67 | 0.65 | 0.63 | 0.62 |
| Molmo2 (Clark et al., 2026) | 4B | 0.83 | 0.61 | 0.77 | 0.62 | 0.60 | 0.59 | 0.60 | 0.48 | 0.47 | 0.47 |
| LLaVA-NeXT-Video (Zhang et al., 2024b) | 7B | 0.62 | 0.45 | 0.65 | 0.48 | 0.33 | 0.33 | 0.33 | 0.30 | 0.28 | 0.29 |
| VideoChat-Flash (Li et al., 2024c) | 7B | 0.94 | 0.40 | 0.90 | 0.36 | 0.58 | 0.58 | 0.58 | 0.49 | 0.49 | 0.49 |
| VideoLLaMA3 (Zhang et al., 2025) | 7B | 0.77 | 0.57 | 0.71 | 0.59 | 0.55 | 0.55 | 0.55 | 0.47 | 0.46 | 0.46 |
| Qwen2.5-VL-Instruct (Bai et al., 2025b) | 7B | 0.64 | 0.68 | 0.56 | 0.71 | 0.57 | 0.55 | 0.55 | 0.51 | 0.49 | 0.49 |
| InternVL3.5 (Wang et al., 2025) | 8B | 0.71 | 0.65 | 0.67 | 0.68 | 0.52 | 0.52 | 0.52 | 0.49 | 0.48 | 0.48 |
| MiniCPM-V-4.5 (Yu et al., 2025) | 8B | 0.77 | 0.64 | 0.80 | 0.62 | 0.60 | 0.59 | 0.59 | 0.51 | 0.51 | 0.51 |
| Qwen3-VL-Instruct (Bai et al., 2025a) | 8B | 0.63 | 0.77 | 0.49 | 0.76 | 0.62 | 0.60 | 0.60 | 0.57 | 0.55 | 0.55 |
| Qwen3-VL-Thinking (Bai et al., 2025a) | 8B | 0.67 | 0.68 | 0.54 | 0.78 | 0.66 | 0.64 | 0.64 | 0.65 | 0.64 | 0.63 |
| Molmo2 (Clark et al., 2026) | 8B | 0.81 | 0.63 | 0.77 | 0.64 | 0.61 | 0.60 | 0.60 | 0.54 | 0.53 | 0.53 |
| Kimi-VL-Instruct (Team et al., 2025) | 16B | 0.69 | 0.70 | 0.50 | 0.76 | 0.61 | 0.59 | 0.59 | 0.55 | 0.54 | 0.53 |
| Kimi-VL-Thinking (Team et al., 2025) | 16B | 0.65 | 0.79 | 0.55 | 0.79 | 0.65 | 0.64 | 0.63 | 0.64 | 0.62 | 0.62 |
| InternVL3.5 (Wang et al., 2025) | 30B | 0.86 | 0.54 | 0.78 | 0.58 | 0.60 | 0.59 | 0.59 | 0.54 | 0.53 | 0.53 |
| Qwen2.5-VL-Instruct (Bai et al., 2025b) | 32B | 0.65 | 0.72 | 0.56 | 0.78 | 0.55 | 0.54 | 0.54 | 0.52 | 0.50 | 0.49 |
| Qwen3-VL-Instruct (Bai et al., 2025a) | 32B | 0.72 | 0.73 | 0.58 | 0.78 | 0.67 | 0.66 | 0.66 | 0.64 | 0.62 | 0.62 |
| Qwen3-VL-Thinking (Bai et al., 2025a) | 32B | 0.69 | 0.76 | 0.48 | 0.81 | 0.75 | 0.72 | 0.72 | 0.72 | 0.70 | 0.70 |
| LLaVA-NeXT-Video (Zhang et al., 2024b) | 34B | 0.65 | 0.47 | 0.68 | 0.50 | 0.37 | 0.38 | 0.37 | 0.34 | 0.32 | 0.33 |
| GLM-4.5v (Hong et al., 2025) | 108B | 0.76 | 0.26 | 0.74 | 0.48 | 0.47 | 0.41 | 0.37 | 0.44 | 0.39 | 0.36 |
| GLM-4.6v-flash (Hong et al., 2025) | 108B | 0.87 | 0.13 | 0.81 | 0.32 | 0.43 | 0.41 | 0.39 | 0.38 | 0.35 | 0.34 |
| GLM-4.6v (Hong et al., 2025) | 108B | 0.81 | 0.22 | 0.79 | 0.36 | 0.50 | 0.46 | 0.44 | 0.55 | 0.50 | 0.48 |
| Qwen3-VL-Instruct (Bai et al., 2025a) | 235B | 0.74 | 0.76 | 0.60 | 0.81 | 0.70 | 0.68 | 0.68 | 0.66 | 0.64 | 0.64 |
| Qwen3-VL-Thinking (Bai et al., 2025a) | 235B | 0.71 | 0.78 | 0.55 | 0.80 | 0.77 | 0.74 | 0.74 | 0.75 | 0.73 | 0.72 |
| *Proprietary Models* | | | | | | | | | | | |
| GPT-4o (OpenAI, a) | - | 0.62 | 0.80 | 0.58 | 0.76 | 0.72 | 0.71 | 0.71 | 0.67 | 0.66 | 0.66 |
| GPT-5 (OpenAI, b) | - | 0.80 | 0.67 | 0.58 | 0.80 | 0.71 | 0.70 | 0.70 | 0.74 | 0.72 | 0.72 |
| GPT-5.2 (OpenAI, c) | - | 0.69 | 0.77 | 0.57 | 0.81 | 0.79 | 0.77 | 0.77 | 0.75 | 0.72 | 0.72 |
| Gemini-2.5-flash (DeepMind, a) | - | 0.72 | 0.75 | 0.53 | 0.79 | 0.73 | 0.70 | 0.70 | 0.71 | 0.69 | 0.68 |
| Gemini-2.5-pro (DeepMind, b) | - | 0.73 | 0.74 | 0.54 | 0.78 | 0.74 | 0.70 | 0.70 | 0.71 | 0.68 | 0.68 |
| Gemini-3-pro (DeepMind, c) | - | 0.68 | 0.84 | 0.53 | 0.84 | 0.80 | 0.79 | 0.79 | 0.75 | 0.74 | 0.74 |
| Doubao-Seed-1.6 (Volcengine, a) | - | 0.80 | 0.69 | 0.56 | 0.79 | 0.72 | 0.71 | 0.71 | 0.73 | 0.71 | 0.71 |
| Doubao-Seed-1.8 (Volcengine, b) | - | 0.77 | 0.76 | 0.55 | 0.80 | 0.79 | 0.76 | 0.76 | 0.73 | 0.67 | 0.67 |

*Table 19.* Main results on other metrics under the AI-generated scenario.

| Model | Size | S_YNQA | | C_YNQA | | S_MCQA | | | C_MCQA | | |
|---|---|---|---|---|---|---|---|---|---|---|---|
| | | YesAcc | NoAcc | YesAcc | NoAcc | Macro Precision | Macro Recall | Macro F1 | Macro Precision | Macro Recall | Macro F1 |
| *Open-source Models* | | | | | | | | | | | |
| VideoChat-Flash (Li et al., 2024c) | 2B | 0.91 | 0.47 | 0.91 | 0.38 | 0.55 | 0.54 | 0.54 | 0.50 | 0.48 | 0.47 |
| VideoLLaMA3 (Zhang et al., 2025) | 2B | 0.69 | 0.75 | 0.69 | 0.65 | 0.51 | 0.35 | 0.25 | 0.42 | 0.32 | 0.20 |
| Qwen3-VL-Instruct (Bai et al., 2025a) | 2B | 0.81 | 0.72 | 0.73 | 0.70 | 0.61 | 0.58 | 0.58 | 0.56 | 0.53 | 0.52 |
| Qwen3-VL-Thinking (Bai et al., 2025a) | 2B | 0.74 | 0.75 | 0.75 | 0.71 | 0.65 | 0.63 | 0.63 | 0.54 | 0.56 | 0.59 |
| Qwen2.5-VL-Instruct (Bai et al., 2025b) | 3B | 0.90 | 0.55 | 0.78 | 0.55 | 0.63 | 0.55 | 0.54 | 0.53 | 0.48 | 0.46 |
| InternVL3.5 (Wang et al., 2025) | 4B | 0.67 | 0.79 | 0.68 | 0.75 | 0.57 | 0.56 | 0.56 | 0.54 | 0.53 | 0.53 |
| MiniCPM-V-4 (Yu et al., 2025) | 4B | 0.87 | 0.61 | 0.83 | 0.60 | 0.54 | 0.54 | 0.54 | 0.49 | 0.47 | 0.46 |
| Qwen3-VL-Instruct (Bai et al., 2025a) | 4B | 0.70 | 0.88 | 0.62 | 0.85 | 0.64 | 0.64 | 0.64 | 0.59 | 0.59 | 0.58 |
| Qwen3-VL-Thinking (Bai et al., 2025a) | 4B | 0.72 | 0.84 | 0.63 | 0.80 | 0.70 | 0.69 | 0.68 | 0.67 | 0.65 | 0.63 |
| Molmo2 (Clark et al., 2026) | 4B | 0.80 | 0.76 | 0.73 | 0.71 | 0.60 | 0.59 | 0.59 | 0.52 | 0.51 | 0.50 |
| LLaVA-NeXT-Video (Zhang et al., 2024b) | 7B | 0.67 | 0.50 | 0.61 | 0.47 | 0.42 | 0.42 | 0.42 | 0.38 | 0.36 | 0.35 |
| VideoChat-Flash (Li et al., 2024c) | 7B | 0.95 | 0.41 | 0.91 | 0.40 | 0.62 | 0.62 | 0.62 | 0.56 | 0.54 | 0.54 |
| VideoLLaMA3 (Zhang et al., 2025) | 7B | 0.79 | 0.72 | 0.76 | 0.66 | 0.59 | 0.57 | 0.57 | 0.49 | 0.48 | 0.46 |
| Qwen2.5-VL-Instruct (Bai et al., 2025b) | 7B | 0.67 | 0.83 | 0.71 | 0.78 | 0.61 | 0.59 | 0.59 | 0.54 | 0.52 | 0.51 |
| InternVL3.5 (Wang et al., 2025) | 8B | 0.71 | 0.77 | 0.76 | 0.71 | 0.57 | 0.57 | 0.56 | 0.53 | 0.53 | 0.51 |
| MiniCPM-V-4.5 (Yu et al., 2025) | 8B | 0.83 | 0.72 | 0.83 | 0.62 | 0.62 | 0.62 | 0.62 | 0.56 | 0.55 | 0.54 |
| Qwen3-VL-Instruct (Bai et al., 2025a) | 8B | 0.69 | 0.91 | 0.61 | 0.86 | 0.68 | 0.66 | 0.66 | 0.63 | 0.61 | 0.60 |
| Qwen3-VL-Thinking (Bai et al., 2025a) | 8B | 0.76 | 0.81 | 0.68 | 0.80 | 0.72 | 0.70 | 0.70 | 0.67 | 0.65 | 0.64 |
| Molmo2 (Clark et al., 2026) | 8B | 0.79 | 0.75 | 0.75 | 0.72 | 0.63 | 0.62 | 0.62 | 0.57 | 0.56 | 0.55 |
| Kimi-VL-Instruct (Team et al., 2025) | 16B | 0.70 | 0.88 | 0.63 | 0.86 | 0.63 | 0.63 | 0.63 | 0.58 | 0.57 | 0.56 |
| Kimi-VL-Thinking (Team et al., 2025) | 16B | 0.72 | 0.84 | 0.64 | 0.81 | 0.69 | 0.68 | 0.68 | 0.65 | 0.63 | 0.62 |
| InternVL3.5 (Wang et al., 2025) | 30B | 0.82 | 0.70 | 0.84 | 0.62 | 0.60 | 0.60 | 0.60 | 0.55 | 0.55 | 0.54 |
| Qwen2.5-VL-Instruct (Bai et al., 2025b) | 32B | 0.73 | 0.83 | 0.70 | 0.78 | 0.66 | 0.64 | 0.64 | 0.59 | 0.54 | 0.53 |
| Qwen3-VL-Instruct (Bai et al., 2025a) | 32B | 0.77 | 0.88 | 0.69 | 0.82 | 0.71 | 0.69 | 0.69 | 0.67 | 0.64 | 0.64 |
| Qwen3-VL-Thinking (Bai et al., 2025a) | 32B | 0.69 | 0.90 | 0.58 | 0.86 | 0.77 | 0.76 | 0.75 | 0.71 | 0.68 | 0.68 |
| LLaVA-NeXT-Video (Zhang et al., 2024b) | 34B | 0.70 | 0.52 | 0.64 | 0.49 | 0.49 | 0.48 | 0.48 | 0.44 | 0.42 | 0.41 |
| GLM-4.5v (Hong et al., 2025) | 108B | 0.72 | 0.34 | 0.68 | 0.55 | 0.50 | 0.43 | 0.41 | 0.49 | 0.44 | 0.42 |
| GLM-4.6v-flash (Hong et al., 2025) | 108B | 0.82 | 0.26 | 0.76 | 0.39 | 0.51 | 0.47 | 0.46 | 0.49 | 0.47 | 0.45 |
| GLM-4.6v (Hong et al., 2025) | 108B | 0.75 | 0.32 | 0.74 | 0.45 | 0.56 | 0.52 | 0.52 | 0.57 | 0.53 | 0.53 |
| Qwen3-VL-Instruct (Bai et al., 2025a) | 235B | 0.78 | 0.88 | 0.70 | 0.84 | 0.72 | 0.71 | 0.70 | 0.68 | 0.65 | 0.65 |
| Qwen3-VL-Thinking (Bai et al., 2025a) | 235B | 0.77 | 0.90 | 0.69 | 0.81 | 0.78 | 0.77 | 0.77 | 0.71 | 0.68 | 0.68 |
| *Proprietary Models* | | | | | | | | | | | |
| GPT-4o (OpenAI, a) | - | 0.67 | 0.84 | 0.69 | 0.80 | 0.68 | 0.67 | 0.67 | 0.66 | 0.65 | 0.64 |
| GPT-5 (OpenAI, b) | - | 0.79 | 0.83 | 0.60 | 0.85 | 0.71 | 0.69 | 0.69 | 0.67 | 0.66 | 0.65 |
| GPT-5.2 (OpenAI, c) | - | 0.73 | 0.88 | 0.65 | 0.83 | 0.79 | 0.76 | 0.76 | 0.72 | 0.68 | 0.67 |
| Gemini-2.5-flash (DeepMind, a) | - | 0.70 | 0.89 | 0.60 | 0.85 | 0.72 | 0.69 | 0.69 | 0.67 | 0.64 | 0.62 |
| Gemini-2.5-pro (DeepMind, b) | - | 0.69 | 0.89 | 0.58 | 0.84 | 0.72 | 0.69 | 0.69 | 0.68 | 0.64 | 0.62 |
| Gemini-3-pro (DeepMind, c) | - | 0.69 | 0.90 | 0.61 | 0.84 | 0.78 | 0.77 | 0.77 | 0.70 | 0.67 | 0.67 |
| Doubao-Seed-1.6 (Volcengine, a) | - | 0.81 | 0.83 | 0.62 | 0.84 | 0.70 | 0.69 | 0.69 | 0.68 | 0.67 | 0.66 |
| Doubao-Seed-1.8 (Volcengine, b) | - | 0.74 | 0.87 | 0.60 | 0.84 | 0.80 | 0.76 | 0.77 | 0.71 | 0.67 | 0.66 |

