# OpenReview forum: "Learning to Decode Against Compositional Hallucination in Video Multimodal Large Language Models"
_ICML.cc/2026/Conference — ICML 2026 regular_

### Official Review · Reviewer_aGCk · 2026-03-08

**Soundness:** 3
**Presentation:** 3
**Significance:** 3
**Originality:** 3
**Overall Recommendation:** 5
**Confidence:** 1

**Summary:**

This paper proposes a benchmark for both isolated and compositional hallucinations occurring in VLLMs to identify real-world coupled patterns. This benchmark contains fine-grained hallucination types and includes two types of QA with real-world videos and AI-generated videos.
This paper proposes a new method to tackle the hallucination problem via a three-step framework, which includes an adaptive perturbation controller to generate a distribution with noise, an SGE module to fuse spatial and temporal information, and finally leverages calibrated logits to fuse these distributions. Reinforcement learning is used to optimize the model.
The experiments show that the proposed method achieves good results on this new dataset.

**Compliance With Llm Reviewing Policy:**

Affirmed.

**Final Justification:**

Thank you for the detailed rebuttal. The authors have fully addressed my concerns by clarifying the distinction between hallucination and perception or reasoning failures. And they justified the use of RL with citations and supplementary experiments to strengthen the purpose of using RL. Also, the authors  acknowledged limitations regarding real-world settings. These responses improve the paper’s soundness and clarity while reinforcing its originality and practical significance. As my initial score already reflected a strong evaluation, I maintain my original rating therefore.

**Key Questions For Authors:**

- The benchmark evaluates hallucination through QA tasks. How does the dataset ensure that incorrect answers correspond to hallucination rather than perception or reasoning failures? Additional clarification on how hallucination is isolated would strengthen the benchmark design.
-  Is RL necessary? Although the authors claim that APC tool selection is a non-differentiable operation, it would be helpful to further justify that you clarify whether alternative approaches (e.g., straight-through estimators) were considered.

**Limitations:**

- The paper briefly mentions potential positive societal implications, such as contributions to the development of more reliable and trustworthy AI systems. However, the discussion mainly focuses on benefits and does not sufficiently address potential limitations or negative societal impacts. The dataset in this benchmark is carefully selected by humans to provide reliable evidence. However, the real-world videos are always noisy and varied.  Perhaps the discussion on these videos could be the focus of future work.

**Strengths And Weaknesses:**

- Soundness: the motivation of this work is clear, and the ablation experiments on these module are solid.
- Presentation: the tasks of the benchmark are clearly defined. The method overview is clear to acknowladge. And the figures and tables are finely readable.
- Significance: This work is significant in defining a wide range of hallucinations in VLLMs, which can be distinguished from prior work. The proposed framework could potentially be applied to existing VLLMs.
- Originality: The work presents a reasonably original attempt to study compositional hallucinations in VLLMs.

---

> ### Author Rebuttal · Authors · 2026-03-31
>
> We thank the reviewer for recognizing our clear motivation, the rigorous compositional hallucination benchmark, solid ablations, and our framework's potential to enhance existing VLLMs.
>
> **Q1. Hallucination vs. Perception/Reasoning.**
>
> We appreciate the reviewer for the suggestion to strengthen our benchmark design.
>
> - For the difference between hallucinations and perceptual or reasoning errors, please refer to the response in **Q1** of Reviewer KnbC.
> - Regarding how we curate the dataset to distinguish between single and compositional hallucinations, please refer to the response in **W2** of Reviewer KnbC.
>
> **Q2. Is RL Necessary?**
>
> RL is a natural fit for our setting. APC makes discrete Bernoulli tool-selection decisions, while the objective is final QA correctness; REINFORCE is a standard way to optimize such stochastic discrete decisions without train-test mismatch [1-3]. We therefore use it as a lightweight optimizer for adaptive routing and beta fusion, rather than as the main contribution itself. STE is a reasonable alternative, but it relies on a biased surrogate gradient and optimizes a relaxed proxy rather than the actual hard routing used at test time [4-5].
>
> We directly test this point with STE. STE backpropagates through hard selections via the straight-through estimator. RL remains the better performing.
>
> | Model                | Avg Acc | $\uparrow$ % |
> | :------------------- | :-----: | :----------: |
> | Qwen3-VL-Instruct-8B |  0.67   |      -       |
> | +Ours w/ RL          |  0.73   |     9.61     |
> | +Ours w/ STE         |  0.69   |     3.89     |
>
> This ablation directly answers the question above: RL is not the only feasible choice, but it is the most effective one in our setting. We further observe that the RL-trained controller improves five backbones by 9.24%~12.03% and transfers to other hallucination benchmarks (i.e., VideoHallucer and EventHallusion) without retraining with 5.48%\~20.22% gains. Full ablation and transfer results are in [Tabs. 3-5](https://anonymous.4open.science/r/A161). And [Figs. 5-7](https://anonymous.4open.science/r/A161) show representative cases of the learned query-dependent tool-selection behavior.
>
> **L1. Societal Impact and Real-world Noise.**
>
> OmniVCHall uses carefully selected, human-verified videos, so it does not fully cover noisier, longer, and more weakly observable real-world videos. We will revise the paper to state this limitation explicitly, note that benchmark gains should not be over-interpreted as robustness in unconstrained or safety-critical deployment, and highlight longer/noisier real-world video evaluation as future work.
>
> [1] Simple statistical gradient-following algorithms for connectionist reinforcement learning. Machine Learning, 1992.
>
> [2] Policy gradient methods for reinforcement learning with function approximation. NeurIPS, 1999.
>
> [3] Learning by playing solving sparse reward tasks from scratch. ICML, 2018.
>
> [4] Estimating or propagating gradients through stochastic neurons for conditional computation. arXiv, 2013.
>
> [5] Categorical reparameterization with gumbel-softmax. arXiv, 2016.

---

> > ### Author Rebuttal · Reviewer_aGCk · 2026-03-31
> >
> > Thanks for explaining. I will keep my evaluation unchanged.

---

> > > ### Author Response · Authors · 2026-04-01
> > >
> > > Thank you for your time and for carefully reading and considering our response. We sincerely appreciate your positive recognition of our explanation.

---

### Official Review · Reviewer_oCtN · 2026-03-08

**Soundness:** 2
**Presentation:** 2
**Significance:** 2
**Originality:** 2
**Overall Recommendation:** 3
**Confidence:** 4

**Summary:**

Current research on video hallucination mitigation primarily focuses on isolated error types, leaving compositional hallucinations—arising from incorrect reasoning over multiple interacting spatial and temporal factors—largely underexplored. This paper introduces OmniVCHall, a benchmark designed to systematically evaluate both isolated and compositional hallucinations in video multimodal large language models. The authors propose TriCD, a contrastive decoding framework with a triple-pathway calibration mechanism. Experimental results show that TriCD consistently improves performance across two representative backbones, achieving an average accuracy improvement of over 10%.

**Compliance With Llm Reviewing Policy:**

Affirmed.

**Key Questions For Authors:**

see weakness

**Limitations:**

yes

**Strengths And Weaknesses:**

Strengths
This paper introduces OmniVCHall, a benchmark designed to systematically evaluate both isolated and compositional hallucinations in video multimodal large language models.

The authors propose TriCD, a contrastive decoding framework with a triple-pathway calibration mechanism.

Experimental results show that TriCD consistently improves performance across two representative backbones, achieving an average accuracy improvement of over 10%.

Weaknesses

TriCD integrates contrastive decoding, perturbation-based negatives, and saliency-based enhancement, but these elements are proposed in prior work. The main contribution lies in combining them rather than introducing fundamentally new modeling techniques.

The proposed hallucination types have a high overlap with existing works, such as Dr.V: A Hierarchical Perception-Temporal-Cognition Framework to Diagnose Video Hallucination by Fine-grained Spatial-Temporal Grounding.

The methods are only validated on model Qwen3-VL-Instruct-8B and VideoLLaMA3-7B.

Although the benchmark contains over 9k QA samples, the video set only consists of 823 videos.

The comparison with baselines is not fair because the proposed method is a training-based methods but the baselines are training-free methods.

TriCD introduces multiple decoding passes and additional modules such as saliency extraction and perturbation selection, which may increase inference latency.

---

> ### Author Rebuttal · Authors · 2026-03-31
>
> We sincerely thank the reviewer for recognizing the value of both our benchmark and method. We have addressed your questions and comments below.
>
> **W1. Beyond Simple Combination.**
>
> TriCD is, to the best of our knowledge, the first learnable adaptive calibration framework specifically designed for compositional hallucinations. TriCD is not a direct stack of existing methods.
>
> - APC: Performs video-query conditioned routing that dynamically adjusts not only the types but also the number of negative perturbations, fundamentally differing from fixed negative branches in conventional contrastive decoding methods
> - SGE: Leverages an explicit gating mechanism to adaptively inject grounded spatio-temporal evidence directly into the decoding process.
>
> The following result shows that direct stacking of prior components improves performance only modestly, while TriCD achieves a much larger gain. This supports that the improvement comes from coordinated triple-path calibration rather than naive combination. Full results are reported in [Tab. 8](https://anonymous.4open.science/r/A161).
>
> | Model                 | Avg  | $\uparrow$ % |
> | :-------------------- | :--: | :----------: |
> | Backbone Avg.         | 0.63 |      -       |
> | +MotionCD & DINO-HEAL | 0.65 |     3.04     |
> | +TCD & DINO-HEAL      | 0.66 |     4.50     |
> | +Ours                 | 0.70 |    10.75     |
>
> **W2. Different with Dr.V.**
>
> A foundational overlap in basic visual error categories (e.g., objects, actions) is inherent and universally expected across the field. The true distinction lies in how the benchmarks are structured and utilized. Dr.V and OmniVCHall are organized along different axes.
>
> - Dr.V is a hierarchical diagnosis framework (perceptive/temporal/cognitive) for identifying where hallucination arises.
> - OmniVCHall is an evidence-type taxonomy for defining what evidence the answer depends on.
>
> Despite some similar labels, their roles fundamentally differ. Unlike Dr.V, we explicitly tackle compositional hallucinations (where multiple evidence types jointly determine correctness) and include a distinct 'camera' type. We also introduce adversarial options such as 'all of the above' and 'none of the above' to make shortcut-based answering harder and better expose compositional errors. Thus, the two works are complementary rather than overlapping.
>
> **W3. Validation on More Backbones.**
>
> We conducted additional experiments across more architectures and tasks. Please refer to the response in **W3** of Reviewer KnbC.
>
> **W4. Dataset Scale and Diversity.**
>
> We conducted further data expansion and explicitly elucidated the core characteristics of OmniVCHall. Please refer to the response in **W1/Q1** of Reviewer HaT6.
>
> **W5. Fairness of Comparison.**
>
> As the first systematic study on compositional hallucinations in VLLMs, we found that existing training-free contrastive decoding methods struggle because they rely on predefined, fixed negative perturbations. However, compositional errors are dynamically driven by complex video-question interactions, making an adaptive, learnable approach essential. We proposed TriCD, the first learnable inference time calibration framework specifically designed for combinatorial illusions. Our TriCD achieves an average improvement of over 10% across five backbones.
>
> - APC learns to route the optimal type and number of negative perturbations.
> - SGE learns to adaptively gate grounded spatio-temporal evidence.
>
> | Method     | Avg  | $\uparrow$ % |
> | :--------- | :--: | :----------: |
> | Backbone   | 0.58 |      -       |
> | +MotionCD  | 0.60 |     2.48     |
> | +TCD       | 0.60 |     3.27     |
> | +DINO-HEAL | 0.60 |     2.82     |
> | +Ours      | 0.64 |    10.35     |
>
> **W6. Inference Latency.**
>
> TriCD's modular design (APC for negative suppression and SGE for positive grounding) is deliberate for tackling complex hallucinations. And TriCD offers an exceptional cost-benefit trade-off. Please refer to the related discussion in **W2** of Reviewer HaT6.

---

> > ### Author Rebuttal · Reviewer_oCtN · 2026-04-05
> >
> > keep my score because of novelty

---

> > > ### Author Response · Authors · 2026-04-05
> > >
> > > We sincerely thank the reviewer for your continued engagement and the rigorous evaluation. We deeply respect your perspective. However, we would like to respectfully clarify the core novelty of our work.
> > >
> > > Our core contribution lies in formally defining a challenging new problem (compositional hallucinations) in video understanding, constructing a comprehensive benchmark to rigorously evaluate it, and developing a targeted, adaptive framework to solve it. Specifically, we advance the field on three distinct fronts:
> > >
> > > 1. **Conceptual Novelty: Discovering the "Compositional Hallucination" Problem**
> > >
> > >   **We are the first to systematically identify and investigate compositional hallucinations in the field of video understanding**, where multiple hallucination types concurrently manifest within a single query. Our extensive evaluation reveals that **39 current VLLMs** suffer a pronounced average accuracy drop of **5.71%** in Yes/No tasks and **9.32%** in multiple-choice tasks when transitioning from single-type to compositional hallucination queries. This empirically proves that while models may handle isolated errors, their reasoning ability significantly decreases under multi-factor compositional complexity. **By exposing this critical vulnerability, our work provides a foundational stepping stone for the community to shift focus toward developing more reliable, reasoning-robust VLLMs.**
> > >
> > > 2. **Benchmark Novelty: Constructing the Comprehensive OmniVCHall**
> > >
> > >   To **rigorously evaluate both isolated and compositional failures**, we constructed OmniVCHall. It is meticulously designed with a highly systematic structure:
> > >
> > >   - **The Novel "Camera" Dimension:** **We are the first to explicitly introduce a camera-based hallucination type** to disentangle cinematic lens dynamics (e.g., zooming/panning) from physical object motion, addressing a profound blind spot in existing studies.
> > >   - **Adversarial Option Injection:** **We are the first to dynamically inject adversarial options** (e.g., "All others are correct" or "None of the above") into video hallucination evaluation. This actively discourages shortcut reasoning and language-only priors, forcing genuine fine-grained cross-modal verification.
> > >   - **Domain Diversity:** We encompass **both authentic real-world dynamics and complex AI-generated artifacts** to ensure a rigorous evaluation environment.
> > >   - **Hierarchical Question Complexity:** Unlike existing datasets that randomly mix queries, **OmniVCHall explicitly segregates evaluation into Single (S) and Compositional (C) tasks.** This deliberate structural axis allows us to precisely quantify the "compositional degradation gap," isolating fundamental multi-factor reasoning bottlenecks from basic perceptual errors.
> > >
> > > 3. **Methodological Novelty: Designing the Adaptive TriCD Framework**
> > >
> > >   Because compositional errors are dynamically driven by complex video-question interactions, conventional contrastive decoding methods relying on static, predefined perturbations fail. **To specifically target the compositional nature of these hallucinations, we propose TriCD, the first triple-pathway adaptive calibration framework optimized via reinforcement learning.**
> > >
> > >   - **Adaptive Perturbation Controller (APC):** Unlike fixed negative branches, APC acts as a dynamic routing network that learns to select context-aware video suppression operations based on the specific compositional complexity of each query.
> > >   - **Saliency-Guided Enhancement (SGE):** It adaptively reinforces grounded token-wise visual evidence by fusing spatial and temporal motion cues directly into the decoding process.
> > >
> > >   **Empirically, this dynamic design enables TriCD to achieve an average improvement of over 10% across five VLLM backbones.** This significantly outperforms naive combinations of static CD methods (which only yield marginal gains of ~3%), proving the necessity and effectiveness of our adaptive architecture.
> > >
> > > We hope this clarifies that our work is a systematic effort to define a critical new problem space, construct a comprehensive benchmark for rigorous evaluation, and introduce a learnable, dynamic paradigm specifically tailored to solve it.

---

### Official Review · Reviewer_HaT6 · 2026-03-13

**Soundness:** 3
**Presentation:** 3
**Significance:** 3
**Originality:** 3
**Overall Recommendation:** 5
**Confidence:** 3

**Summary:**

The paper seeks to focus on a central concept: compositional hallucination in video multimodal large language models (VLLMs), where models fail when reasoning about multiple interacting visual factors such as objects, actions, temporal events, and camera motion. The article's important contribution is the introduction of OmniVCHall, a benchmark designed to evaluate both isolated and compositional hallucinations in VLLMs across eight hallucination categories, along with adversarial question options to prevent shortcut reasoning. The paper also proposes TriCD, a triple-pathway contrastive decoding framework that combines adaptive perturbation-based negative sampling and saliency-guided visual grounding to mitigate hallucinations. Experimental results show that TriCD consistently improves accuracy on several VLLM backbones and particularly benefits complex compositional reasoning tasks.

**Compliance With Llm Reviewing Policy:**

Affirmed.

**Key Questions For Authors:**

1. The benchmark contains 823 videos and 9,027 QA samples, which may still be relatively small for evaluating large multimodal models. It would be helpful to clarify whether the dataset scale is sufficient to represent the diversity of real-world compositional hallucinations.

2. The proposed method relies on external modules such as DINOv3 and optical flow estimation, which introduce additional computation. It would be useful to discuss how the framework scales to long videos or large deployment scenarios.

3. While TriCD improves accuracy on several models, the paper mainly tests two representative backbones. More experiments on a broader set of models would strengthen the claim that the approach generalizes well across architectures.

**Limitations:**

Please refer to the weakness.

**Strengths And Weaknesses:**

Strengths

1. Novel problem formulation. The paper highlights compositional hallucination, which is a realistic but underexplored failure mode in VLLMs. By explicitly studying interactions among multiple visual factors, the work identifies an important gap in current multimodal evaluation and contributes a clearer understanding of model limitations.

2. Comprehensive benchmark design. OmniVCHall introduces eight hallucination categories, including a new camera-based hallucination type, and incorporates adversarial answer options to reduce shortcut reasoning. This design provides a more rigorous diagnostic benchmark compared to previous datasets that mainly focus on single hallucination types.

3. Effective decoding framework. The proposed TriCD framework combines adaptive negative perturbation with saliency-based visual grounding, creating a contrastive decoding process that improves robustness without modifying model parameters. Experimental results demonstrate consistent improvements across multiple tasks, especially for compositional reasoning scenarios.

Weakness

1. Limited dataset scale. Although the benchmark is carefully constructed, the total number of videos (823) and QA pairs (9k) is relatively modest compared with large-scale vision-language benchmarks. This may limit the diversity of scenarios and reduce confidence in the generality of the evaluation results.

2. Method complexity. TriCD introduces several components, including an adaptive perturbation controller, saliency fusion module, and reinforcement learning optimization. This complexity may make the framework harder to reproduce or deploy compared with simpler decoding-based approaches.

3. Dependence on auxiliary vision models. The approach relies on external models such as DINOv3 and optical flow algorithms to compute saliency signals. This dependence may introduce additional computational overhead and potentially limit applicability in real-time video understanding systems.

4. Limited evaluation scope. Although the benchmark evaluates 39 models, the TriCD method itself is tested on only a small number of backbones. Additional experiments across more architectures and tasks would help validate the general applicability of the proposed method.

---

> ### Author Rebuttal · Authors · 2026-03-31
>
> We thank the reviewer for recognizing the novelty of tackling compositional VLLM hallucinations, the rigorous OmniVCHall benchmark, and TriCD's effectiveness.
>
> **W1/Q1. Dataset Scale and Diversity.**
>
> We appreciate the feedback regarding the dataset scale. We have also continued expanding the dataset after the initial submission. As summarized in [Tabs. 9-11](https://anonymous.4open.science/r/A161), the current extension adds 337 videos (171 AI-generated, 166 real-world) and 3,177 QAs while preserving all four sub-tasks. The updated OmniVCHall now totals 1,160 videos and 12,204 QAs, further strengthening its scale without compromising diagnostic quality.
>
> | Domain | Human Activities | Animals | Transportation | Outdoor | Household | Daily Life | Nature | Fantasy | Sum  |
> | :----- | :--------------: | :-----: | :------------: | :-----: | :-------: | :--------: | :----: | :-----: | :--: |
> | Number |       207        |   156   |       68       |   69    |    168    |    174     |  172   |   146   | 1160 |
>
> We have reported some of the model's performance on the new dataset. We will expand the evaluation in the final version.
>
> |    Video     |    Model     | S_YNQA | C_YNQA | S_MCQA | C_MCQA |
> | :----------: | :----------: | :----: | :----: | :----: | :----: |
> | AI-generated |   GPT-5.2    | 0.811  | 0.718  | 0.759  | 0.672  |
> |              | Gemini-3-pro | 0.803  | 0.701  | 0.754  | 0.658  |
> |  Real-world  |   GPT-5.2    | 0.748  | 0.719  | 0.784  | 0.728  |
> |              | Gemini-3-pro | 0.779  | 0.712  | 0.793  | 0.745  |
>
> Referring to the hallucination research literature [1-2], OmniVCHall is a highly competitive diagnostic dataset for compositional hallucination prioritizing diversity and quality. It features:
>
> - Scale within the domain: As shown in Tab. 2 in our original paper, our 9,027 QAs surpass most existing hallucination benchmarks, including TempCompass (7.5K), ELV-Halluc (4.8K), and MHBench (3.6K).
> - Diversity: Existing datasets with larger video counts often sacrifice coverage. For example, VidHalluc (5K videos) lacks 4 crucial hallucination types and contains zero AI-generated videos. In contrast, OmniVCHall maximizes diagnostic richness: it balances real/AI videos; spans 11 heterogeneous sources across 8 domains; covers single/compositional settings via YNQA and MCQA formats; and employs adversarial answer options with fine-grained annotations across 8 evidence types.
>
> **W2. Method Complexity.**
>
> TriCD's modular design (APC for negative suppression and SGE for positive grounding) is deliberate for tackling complex hallucinations.
>
> With rapid advancements in LLM inference acceleration systems (e.g., vLLM, FlashAttention), the community increasingly prioritizes generation quality over strict latency for complex reasoning. This paradigm is evident in frameworks like Tree-of-Thoughts [3], Graph-of-Thoughts [4], and similarly in the contrastive decoding domain with multi-stage methods like SECOND [5].
>
> Aligned with this trend, TriCD offers an exceptional cost-benefit trade-off: it adds only ~1s of extra inference overhead (vs. MotionCD and TCD) yet delivers 5.3x and 2.7x their relative performance gains, respectively. Thus, this moderate overhead vs. substantial performance gain is highly worthwhile.
>
> **W3/Q2. Scalability.**
>
> - Real-time videos: TriCD adds only ~1s of inference overhead (9.12s vs. MotionCD's 8.05s and TCD's 7.95s) yet delivers 5.3x and 2.7x their relative performance gains, respectively. Thus, the overhead is moderate relative to the gain, although strict real-time processing remains challenging.
> - Long-time videos: Saliency extraction and multi-branch decoding scale substantially with frame count, especially for compositional cases where evidence is widely distributed.
>
> To address this, future efficiency optimization will integrate mature strategies like adaptive/query-aware keyframe sampling [6] and sparse-memory compression [7], applying full calibration only to high-risk segments. Full time-performance comparison results are shown in the original Appendix D.
>
> **W4/Q3. More Robust Evaluation.**
>
> We conducted additional experiments across more architectures and tasks. Please refer to the response in **W3** of Reviewer KnbC.
>
> [1] Videohallucer: Evaluating intrinsic and extrinsic hallucinations in large video-language models. arXiv, 2024.
>
> [2] Eventhallusion: Diagnosing event hallucinations in video llms. arXiv, 2024.
>
> [3] Tree of thoughts: Deliberate problem solving with large language models. NeurIPS, 2023.
>
> [4] Graph of thoughts: Solving elaborate problems with large language models. AAAI, 2024.
>
> [5] Second: Mitigating perceptual hallucination in vision-language models via selective and contrastive decoding. ICML, 2025.
>
> [6] Adaframe: Adaptive frame selection for fast video recognition. CVPR, 2019.
>
> [7] Moviechat: From dense token to sparse memory for long video understanding. CVPR, 2024.

---

> > ### Author Rebuttal · Reviewer_HaT6 · 2026-04-03
> >
> > The rebuttal has addressed all my concerns. Thanks for the authors' rebuttal.

---

> > > ### Author Response · Authors · 2026-04-03
> > >
> > > Thank you for your time and thoughtful review of our response. We greatly appreciate your positive feedback on our explanation.

---

### Official Review · Reviewer_KnbC · 2026-03-16

**Soundness:** 3
**Presentation:** 2
**Significance:** 3
**Originality:** 3
**Overall Recommendation:** 3
**Confidence:** 5

**Summary:**

This paper studies hallucination in vLLMs and focuses on compositional hallucination, where errors arise from interactions among multiple semantic factors. The authors argue that existing benchmarks largely evaluate hallucinations in isolation and fail to capture such compositional failure modes.

To address this, the paper introduces OmniVCHall, a benchmark that evaluates hallucinations across 8 semantic categories and supports both single and compositional scenarios. It also proposes TriCD, a triple-pathway contrastive decoding framework that mitigates hallucinations during inference through adaptive perturbations and saliency-guided enhancement without updating model parameters.

Experiments across multiple VLLMs show that TriCD improves performance on the proposed benchmark.

**Compliance With Llm Reviewing Policy:**

Affirmed.

**Final Justification:**

I appreciate the authors’ rebuttals, and I find the idea of compositional hallucination very interesting. However, the paper still lacks sufficient rigor in its current form. The original submission did not clearly define the core concept, and although the rebuttal attempts to clarify it, the definition remains problematic. In addition, several important prior works are still missing from the discussion. Overall, I believe the paper would benefit from substantial revision (potentially including updates to the dataset to better align with a clearer and more robust definition). For these reasons, I cannot recommend acceptance at this time.

**Key Questions For Authors:**

Which compositional hallucinations are simply perception errors, which are not?

**Limitations:**

No.

**Strengths And Weaknesses:**

Strengths:
1. The paper highlights an important limitation of current vLLMs that hallucinations often occur due to compositional reasoning failures, rather than single-factor mistakes.
2. The proposed TriCD approach operates at test time and can be applied to existing models without retraining.
3. The proposed OmniVCHall benchmark is reasonably well structured.

Weaknesses:
1. Missing related works on compositional hallucinations and reasoning, multimodal hallucinations, and video reasoning. Better to have pointers to the related work in the appendix.
2. It is less straightforward to capture the concrete ideas of typical compositional hallucinations. Many cases still appear to be object/action recognition errors. For instance, the example in Fig. 2(c) "What action is the hand performing on the apple?" is, in nature, a single action recognition hallucination. How to clearly define compositional hallucinations strictly and distinguish them from others is less clear.
3. It is less clear the usefulness of the use of RL in this paper, since it is simply REINFORCE + binary rewards.

---

> ### Author Rebuttal · Authors · 2026-03-31
>
> We thank the reviewer for recognizing the value of compositional hallucinations, the test-time utility of TriCD, and our well-structured OmniVCHall benchmark.
>
> **W1. Added Discussion on Related Work.**
>
> We have added a dedicated discussion of related work to the original Appendix A.
>
> **W2. Clarify the Definition of Single and Compositional Hallucination.**
>
> Fig. 2(c) was intended to illustrate adversarial option injection rather than a canonical compositional hallucination case. We now add true compositional examples in [Fig. 1](https://anonymous.4open.science/r/A161/).
>
> To make the distinction explicit, we add a formal definition:
>
> - Let $T(V,Q,A)$ be the minimal evidence-type set needed to support answer $A$ for question $Q$ on video $V$.
> - A case is single if $|T|=1$, and compositional if $|T|\geq2$ and removing any type makes the answer unsupported, ambiguous, or wrong.
>
> This follows the minimal-sufficiency view in explainable AI, where a decision is characterized by the minimal subset of features/reasons sufficient to preserve it [1-2].
>
> A blinded re-annotation audit (3 annotators $\times$ 80 mixed cases each, using only VQA pairs) confirms our definition's reproducibility. High binary accuracy and Jaccard similarity demonstrate it reliably distinguishes single/compositional cases and helps recover minimal evidence-type sets. For the full results and case studies, see [Figs. 2-4 and Tabs. 1-2](https://anonymous.4open.science/r/A161/).
>
> | Metric     |     Overall     |
> | :--------- | :-------------: |
> | BinaryAcc  | 0.9056 ± 0.0049 |
> | JaccardSim | 0.8988 ± 0.0015 |
>
> **W3. RL Usefulness.**
>
> RL is used because APC requires hard tool selection, while the objective is final QA correctness. Under this frozen-backbone, lightweight-controller setting, REINFORCE+$\pm1$ is a simple and direct choice for optimizing discrete Bernoulli decisions and beta fusion.
>
> We compare RL with a simpler alternative. RL remains the best-performing option. Please refer to the response in **Q3** of Reviewer aGCk.
>
> We have extended the evaluation from the original two backbones to five in total, and TriCD with RL consistently improves all of them by 9.24%\~12.03%:
>
> | Backbone             | +**Ours** Avg | $\uparrow$ % |
> | :------------------- | :-----------: | :----------: |
> | Qwen3-VL-Instruct-8B |     0.73      |     9.61     |
> | VideoLLaMA3-7B       |     0.67      |    12.03     |
> | LLaVA-NeXT-Video-7B  |     0.51      |    11.48     |
> | InternVL3.5-8B       |     0.68      |     9.24     |
> | VideoChat-Flash-7B   |     0.62      |     9.76     |
>
> The RL-trained controller improves all five tested backbones by 9.24%\~12.03%, and after training once on OmniVCHall, it transfers to VideoHallucer and EventHallusion with 5.48%\~20.22% gains.
>
> | Method        | Ours | VideoHallucer | EventHallusion | $\uparrow$ % |
> | :------------ | :--: | :-----------: | :------------: | :----------: |
> | Backbone Avg. | 0.58 |     0.42      |      0.59      |      -       |
> | +Ours (RL)    | 0.64 |     0.49      |      0.62      |    10.92     |
>
> Finally, [Figs. 5-7](https://anonymous.4open.science/r/A161/) show representative cases where the RL-trained controller learns meaningful query-dependent tool selection and complementary saliency fusion. Full results are reported in [Tabs. 3-5](https://anonymous.4open.science/r/A161/).
>
> **Q1. Hallucination vs. Perception Errors.**
>
> Hallucination and perception/reasoning errors describe different aspects of model behavior. Hallucination targets the observed output-evidence inconsistency (the resulting outcome), whereas perception and reasoning are its underlying causes [3-4]. Thus, triggering causes should not be conflated with the hallucination itself.
>
> To test this empirically, we decompose each wrong compositional case into its constituent single-type questions. We identify three failure patterns. Full results are in [Figs. 8-9 and Tab. 6](https://anonymous.4open.science/r/A161/).
>
> - All Wrong (minority, 8.0%): All atomic questions fail, proving compositional errors are rarely pure perception failures.
> - All Correct (substantial, 22.5%): All atomic questions pass, but the compositional question still fails, highlighting reasoning/composition-dominant errors.
> - Mixed (majority, 69.4%): Partial atomic failures, indicating most compositional failures stem from coupled perception and reasoning failures.
>
> These findings confirm that compositional hallucinations are complex, coupled phenomena, making their explicit study a critical necessity for advancing VLLMs.
>
> [1] Consistent sufficient explanations and minimal local rules for explaining the decision of any classifier or regressor. NeurIPS, 2022.
>
> [2] On the computation of necessary and sufficient explanations. AAAI, 2022.
>
> [3] Mirage: Assessing hallucination in multimodal reasoning chains of mllm. arXiv, 2025.
>
> [4] Is cognition consistent with perception? assessing and mitigating multimodal knowledge conflicts in document understanding. EMNLP, 2025.

---

> > ### Author Rebuttal · Reviewer_KnbC · 2026-04-04
> >
> > 1. I've checked the related work section, and multiple video-hallucination references are still missing.
> > 2. The definition seems less reasonable to me since there could be compositional hallucinations caused by multiple hallucinations at the same time.

---

> > > ### Author Response · Authors · 2026-04-04
> > >
> > > We sincerely thank the reviewer for the continued engagement and for raising these important follow-up questions. We deeply appreciate your meticulousness, which helps us further clarify our contributions.
> > >
> > > **Reply to Follow-up Q1:**
> > >
> > > We apologize for the confusion regarding the updated related work and sincerely appreciate your careful attention to its completeness.
> > >
> > > Due to the ICML 2026 rebuttal policy, we are unable to revise the manuscript PDF or provide additional anonymous text externally during the rebuttal period. Therefore, we include below the newly added discussion, which will be incorporated into the camera-ready version. For clarity, the new text is italicized and shown with its original surrounding context.
> > >
> > > 1. Additions to A.1.
> > >
> > > > ... perception and efficiency. *Early video reasoning methods focused on learning aligned spatiotemporal representations for basic temporal understanding [1]. More recent work has shifted toward multi-step video reasoning, enabling iterative inference over longer temporal contexts [2-4].* However, current VLLMs ...
> > >
> > > 2. Additions to A.2.
> > >
> > > > ... in a single query. *For image understanding, compositional hallucinations have been actively explored, spanning from the discovery of attribute binding failures [5] and multi-dimensional evaluations [6-7], to targeted mitigations like reminder compositions [8] and LLM-guided extraction [9]. However, directly extending these static paradigms to the video domain introduces profound temporal complexities. In dynamic scenes, compositional errors frequently trigger cascading reasoning failures across time, rather than remaining as isolated static mistakes.* These limitations ...
> > >
> > > 3. Additions to A.3.
> > >
> > > > ... prior linguistic knowledge. *Furthermore, in complex multi-step tasks, these multimodal hallucinations are often exacerbated by phenomena like "visual thinking drift" [2] and intermediate "mirages" [10].* Various strategies have ...
> > >
> > > > ... specialized training objectives. *And, techniques such as dynamically injecting contextual reminders during inference [8] and utilizing external language models to explicitly parse visual structures [9] have been proposed to prevent the semantic unbinding of objects and attributes.*
> > >
> > > **Reply to Follow-up Q2:**
> > >
> > > We thank the reviewer for this important clarification. We agree that compositional hallucinations can involve multiple simultaneous hallucinations. Our definition naturally covers this exact scenario.
> > >
> > > **The key point is that compositionality is defined by the number of required evidence types, rather than the number of errors the model makes. Specifically, a case is compositional if the correct answer depends on multiple interdependent conditions.** For example, a single case (e.g., "What color is the car?", $|T|=1$) evaluates the attribute. In contrast, a compositional case (e.g., "Is the red car turning left?", $|T| \ge 2$) requires binding multiple types. In the latter, the reasoning fails regardless of whether the model commits a single error (e.g., hallucinating the color as blue) or multiple concurrent errors (e.g., hallucinating both the color as blue and the action as turning right). Therefore, failures may arise from hallucinations on one, several, or all of these conditions, but they all correspond to the same compositional setting.
> > >
> > > Empirically, our atomic decomposition (please see our previous response to **Q1**) validates this. The "All Wrong" (8.0%) and "Mixed" (69.4%) patterns demonstrate that 77.4% of compositional failures are driven exactly by multiple concurrent hallucinations. The remaining "All Correct" (22.6%) cases represent pure compositional reasoning failures, where individual elements are perceived correctly but their combination is not. Importantly, these patterns describe how errors manifest, rather than redefining compositionality itself.
> > >
> > > We hope our response has addressed your concerns and would be grateful if you could consider raising your score.
> > >
> > > [1] Self-chained image-language model for video localization and question answering. NeurIPS, 2023.
> > >
> > > [2] When thinking drifts: Evidential grounding for robust video reasoning. NeurIPS, 2025.
> > >
> > > [3] Alternating perception-reasoning for hallucination-resistant video understanding. arXiv, 2025.
> > >
> > > [4] Divide and conquer: Exploring language-centric tree reasoning for video question-answering. ICML, 2025.
> > >
> > > [5] Discovering compositional hallucinations in lvlms. NeurIPS, 2025.
> > >
> > > [6] GHOST: Getting to the bottom of hallucinations with a multi-round consistency benchmark. WACV, 2026.
> > >
> > > [7] Evaluating image hallucination in text-to-image generation with question-answering. AAAI, 2025.
> > >
> > > [8] ReCo: Reminder composition mitigates hallucinations in vision-language models. arXiv, 2025.
> > >
> > >  [9] Seeing beyond hallucinations: Llm-based compositional information extraction for multimodal reasoning. ACM SIGIR, 2025.
> > >
> > > [10] Mirage: Assessing hallucination in multimodal reasoning chains of mllm. arXiv, 2025.

---

### Decision · Program_Chairs · 2026-04-30

**Decision:**

Accept (regular)

**Comment:**

Summaries:
This paper introduces OmniVCHall, a benchmark to evaluate isolated and compositional hallucinations in video multimodal large language models. OmniVCHall covers 8 video hallucination types and was evaluated on 39 VLMs, showing a clear gap with human performance. To mitigate these hallucinations, the authors propose TriCD, a contrastive decoding framework with a triple-pathway calibration mechanism. The experiments show that TriCD achieves an average 10% improvement in accuracy over the Qwen3-VL-Instruct-8B and VideoLlaMA-3-7B baselines.

Justifications:
A comprehensive evaluation of 39 VLMs reveals a clear gap between the proposed compositional hallucination benchmark and human performance. The rebuttal fully addressed the concerns raised by reviewers HaT6 and aGCk, leading to a solid score (5 - accept). Reviewer KnbC also marked the rebuttal as partially resolved, but kept the original score (3 - weak reject), which has not been a critical weakness (missing related work, which was addressed in the follow-up rebuttal). The RL concerns were being addressed by the authors. Reviewer oCtN did not respond to the authors' comments during the author-reviewer period, so I consider a lower weight of this review score.